# Modified rice bran arabinoxylan as a nutraceutical in health and disease—A scoping review with bibliometric analysis

**Soo Liang Ooi**[1], **Peter S. Micalos**[2], **Sok Cheon Pak**[1] *

1 School of Dentistry and Medical Sciences, Charles Sturt University, Bathurst, New South Wales, Australia,
2 School of Dentistry and Medical Sciences, Charles Sturt University, Port Macquarie, New South Wales Australia

* spak@csu.edu.au

## Abstract

Rice bran arabinoxylan compound (RBAC) is a polysaccharide modified by *Lentinus edodes* mycelial enzyme widely used as a nutraceutical. To explore translational research on RBAC, a scoping review was conducted to synthesise research evidence from English (MEDLINE, ProQuest, CENTRAL, Emcare, CINAHL+, Web of Science), Japanese (CiNii, J-Stage), Korean (KCI, RISS, ScienceON), and Chinese (CNKI, Wanfang) sources while combining bibliometrics and network analyses for data visualisation. Searches were conducted between September and October 2022. Ninety-eight articles on RBAC and the biological activities related to human health or disease were included. Research progressed with linear growth (median = 3/year) from 1998 to 2022, predominantly on Biobran MGN-3 (86.73%) and contributed by 289 authors from 100 institutions across 18 countries. Clinical studies constitute 61.1% of recent articles (2018 to 2022). Over 50% of the research was from the USA (29/98, 29.59%) and Japan (22/98, 22.45%). A shifting focus from immuno-cellular activities to human translations over the years was shown via keyword visualisation. Beneficial effects of RBAC include immunomodulation, synergistic anticancer properties, hepatoprotection, antiinflammation, and antioxidation. As an oral supplement taken as an adjuvant during chemoradiotherapy, cancer patients reported reduced side effects and improved quality of life in human studies, indicating RBAC's impact on the psycho-neuro-immune axis. RBAC has been studied in 17 conditions, including cancer, liver diseases, HIV, allergy, chronic fatigue, gastroenteritis, cold/flu, diabetes, and in healthy participants. Further translational research on the impact on patient and community health is required for the evidence-informed use of RBAC in health and disease.

## Introduction

Rice bran is a by-product of rice milling. In 2021, the global rice production was 787 million metric tons, with approximately 80 million tons of rice bran being generated [1]. Traditionally, rice bran was considered an agricultural waste with only limited usage as animal feed [2]. Disposing such a high volume of industrial food residuals is a substantial undertaking and,

**Data Availability Statement:** The data underlying the results presented in the study are available from https://github.com/sooi10/RBACScoping.

**Funding:** S.C.P. received funding from Charles Sturt University under the Tri-faculty Open Access

Publishing Scheme 2023 for payment of open-acess publication fees. The funder had no role in study design, data collection and analysis, decision to publish, or preparation of the manuscript.

**Competing interests:** The authors were assisted by Mr Ryo Ninomiya from Daiwa Pharmaceutical Co., Ltd. (Tokyo, Japan) and Dr Seong Gil Hong from Erom Co., Ltd. (Chuncheon, South Korea) in searching and sourcing selected Japanese and Korean articles, respectively. These individuals were employees of commercial companies that produced and marketed RBAC products. However, both individuals and their affiliated organisations had no role in the implementation, analyses, data interpretation, or decision to submit the study results. The authors have declared that no other competing interests exist. This does not alter our adherence to PLOS ONE policies on sharing data and materials.

without proper management, can pose significant environmental and public health issues [3]. Consequently, much research has been devoted to finding sustainable ways to recycle and reuse rice bran, most notably as a functional and nutritional food ingredient, by leveraging on the many bioactive compounds, such as phenolics, tocotrienols, tocopherols, and γ-oryzanol [2]. Moreover, rice bran is also a rich source of dietary fibre, comprising 8.5% arabinoxylans, which acts as a prebiotic for promoting gut health through stimulating beneficial gut microbial species to improve immune health and prevent chronic disease [4, 5].

Rice bran arabinoxylan compound (RBAC) is the generic name for any arabinoxylan-rich derivative of rice bran enzymatically modified with *Lentinus edodes* mycelium. The production of RBAC typically involves the preparation of rice bran in a growth medium with sterilisation, followed by bioconversion through fermentation with *L. edodes* enzyme over a period of time before extraction, purification, and drying of the compound into a water-soluble powder [6]. RBAC products are commercially manufactured and marketed as nutraceuticals for enhancing immune functions [6]. Biobran MGN-3, first developed by Daiwa Pharmaceutical Co., Ltd. (Tokyo, Japan; hereafter referred to as Daiwa), is the most well-known RBAC worldwide [7]. Rice bran exo-biopolymer (RBEP), developed by Erom Co., Ltd. (Chuncheon, South Korea; hereafter referred to as Erom) and used as the main ingredient in immune-related functional food and nutraceutical products, is another derivative cited in literature [8–10].

Research on RBAC as an immune-modulating substance was first pioneered by Ghoneum [11, 12]. Over the years, there has been a growing interest in translating the in vivo and in vitro biological activities demonstrated by RBAC to treatment effects in humans for clinical applications. Translational research refers to the steps from 'the bench to the bedside' that incorporate promising scientific findings into evidence-based practice and impact the community [13]. RBAC's most notable clinical application is the potential in cancer immunotherapy as an adjuvant oral supplement [14]. RBAC has also been promoted as an immune-modulating functional food for chronic inflammatory conditions such as human immunodeficiency virus (HIV), diabetes, hepatitis, arthritis, and chronic fatigue, to name a few [15–18].

Previous reviews have assessed the available evidence on RBAC as a complementary therapy for conventional cancer treatment [14, 19–21] and explored the health-promoting properties for the potential clinical application in chronic conditions [6, 22–24]. Despite these efforts, the synthesis of available scientific research on RBAC remains insufficient. Previous reviews were based almost exclusively on English-language publications. Studies conducted in Japan and South Korea on RBAC may not be available in English. Also, with the growing interest in natural products for health and wellness in China in recent decades [25], parallel discoveries and research on RBAC may be reported in Chinese literature. Therefore, this review aims to conduct inclusive systematic searches that map the available studies on RBAC, from basic research to human studies in English, Japanese, Korean, and Chinese literature. A scoping study is the most appropriate research synthesis method for the potentially broad and diverse scientific literature.

The initial scoping question on RBAC was: 'What is known about the beneficial effects of RBAC on health and disease conditions from basic research to human studies?' The question was intentionally broad at the onset of the study and has been subsequently revised post hoc to 'What is known about the translational research on RBAC and its potential beneficial effects on health and disease conditions?' The revision enabled the reviewers to incorporate a bibliometric analysis of all the available literature based on the authors, institutions, publications, and research impact to understand the translational progress of RBAC over time.

## Materials and methods

### Protocol and registration

This scoping review followed the JBI methodology for scoping reviews [26]. A preliminary search on Pubmed (MEDLINE), the Cochrane Database of Systematic Reviews and JBI Evidence Synthesis found no similar systematic or scoping reviews. Before commencing the search, a study protocol was registered on OSF Registries (DOI:10.17605/OSF.IO/5PQ8W) for public access [27].

### Eligibility criteria

RBAC is a product class of any rice bran arabinoxylan/polysaccharide extract produced through bioconversion with *L. edodes* mycelial enzyme. Studies on arabinoxylans or polysaccharides extracted from other cereal grains, unmodified rice bran, and other rice bran derivatives produced without using *L. edodes* mycelial enzyme were excluded. To ensure that only studies on RBAC were selected, an article must contain either (1) an explanation of or references to how the extraction was performed or (2) the brand/product name or provider of the RBAC used. Failure to do so would result in exclusion.

This scoping review considered only scholarly articles reporting results from primary research on the effects of RBAC on biological activities related to human health or disease conditions. Opinion papers and non-academic sources (e.g., magazines, news articles, and trade journals) were excluded. Any secondary research based on existing data, such as a systematic or narrative review, was excluded unless it contained a secondary data analysis that reported previously unknown findings. Conference presentations and posters were not considered; however, conference abstracts with the results not published in full-text articles were included. Clinical trial registrations and study protocols were included for publication rate assessment only. The World Health Organization [28] recommends all interventional trials be registered on a public registry to prevent publication bias and selective reporting. However, mandatory registration is only a requirement since 2007. Furthermore, only clinical trials beyond Phase 1 involving regulated pharmaceutical products and devices must be registered on ClinicalTrial. gov in the USA [29]. Hence, not all interventional studies included in this review were subjected to registration, especially those with single-arm design with no control. The publication rate assessment will be based only on controlled human trials after 2007.

When considering human studies, there were no exclusion criteria for participants, such as age, sex, geographical location, health status, or disease conditions. However, RBAC was to be used as a nutraceutical for human consumption. Thus, only oral administration was allowed in all human clinical studies. Other routes of administration, such as intraperitoneal injection, were permitted in animal studies for the understanding of its underlying mechanisms. This review also included all study designs covering interventional (experimental and quasi-experimental) and observational (case series, individual case reports and descriptive cross-sectional) studies.

This review also considered preclinical studies, including in vivo experiments using cell cultures on RBAC and animal studies with models that mimic aspects of physiological processes or human diseases. Non-human studies investigating RBAC for manufacturing, agriculture, or other purposes unrelated to human health and conditions were excluded.

### Information sources

This review aimed to source both published and unpublished studies. A total of 13 academic databases were used in the systematic searches, consisting of six international databases in

English, two Chinese databases, three Korean databases, and two Japanese databases. The complete list of these databases is summarised in Table 1. It is recognised that the inclusion of grey literature could reduce publication bias, increase the comprehensiveness and timeliness of a systematic review, and foster a balanced picture of available evidence [30]. Hence, to uncover unpublished studies/grey literature, the reviewers searched the official website of Biobran MGN-3 [31], collected papers on Biobran MGN-3 compiled by BioBran Research Foundation [32], and the Institute for Progressive Research of RBAC Immunomodulator Compounds website [33]. The references of all included articles and review papers on RBAC were also screened for additional studies.

## Search strategy

An initial limited search of MEDLINE (via PubMed) and CINAHL was undertaken to identify articles on the topic. The words in the titles and abstracts of relevant articles and the index terms used to describe the papers were used to develop a complete search strategy without restricting the language and publication date. The search strategy, including all identified keywords and index terms, was then adapted for each database or information source. The search strategy for MEDLINE (via PubMed) is available as an example in S1.1 in S1 File.

## Source of evidence selection

Following the search, all identified records were collated and uploaded into EndNote 20 (Clarivate Analytics, PA, USA) with duplicates removed. Titles and abstracts were screened by one reviewer (SLO) for assessment against the inclusion criteria and verified by another reviewer (SCP) after a pilot test. Potentially relevant sources were retrieved, and their full-text files were imported into EndNote 20 for management, assessment and review of information. The full text of each selected record was assessed in detail against the inclusion criteria by one reviewer (SLO) and verified by at least one other team member (SCP or PSM). Reasons for excluding any full-text articles were recorded and reported in the scoping review. Any disagreements during the selection process were resolved through discussion to reach a consensus.

## Data extraction

Data were extracted from included articles by a reviewer (SLO) and verified by another (SCP) using a data extraction form developed by the reviewers (See S1.2 in S1 File). Non-English language articles were first translated to English using Google Translate (Google, Mountain View, CA, USA) and then revised and edited by one of the reviewers familiar with the source language. The translated versions were then used for data extraction. Many Japanese articles were

**Table 1. Information sources for published studies.**

| Database Type | Count | List |
|---|---|---|
| English databases | 6 | MEDLINE (via PubMed), ProQuest, Cochrane Register of Controlled Trials (CENTRAL), Emcare (via Ovid), Cumulative Index to Nursing & Allied Health Literature (CINAHL plus, via EBSCO), and Web of Science (exclude MEDLINE & KCI). |
| Chinese databases | 2 | Chinese National Knowledge Infrastructure Database (CNKI) and Wanfang. |
| Korean databases | 3 | Korean Journal Database (KCI, via Web of Science), Research Information Service System (RISS), and ScienceON. |
| Japanese databases | 2 | CiNii and J-Stage. |

professionally translated into English for public access on the Biobran.org website. The translated copies of these articles were used in this review where available.

Extracted data included specific details about the article, year of publication, authors, country of origin, study design, participants, concept, context, methodology, outcome measures and key findings relevant to the review question. Citation count is a measure of research impact. The reviewers also searched Google Scholar (Google, Mountain View, CA, USA) to extract the citation count for each included article. Medical Subject Headings (MeSH) terms were used for standardised keywords analysis. MeSH is a controlled and hierarchically organised vocabulary for biomedical and health-related information. It is maintained by the United States National Library of Medicine for indexing, cataloguing and searching, especially in PubMed and MEDLINE. The reviewers compiled MeSH terms for each included source of evidence from two sources: (1) PubMed and (2) MeSH on Demand based on the article's abstract. Since a published paper has multiple authors whose contributions may not be equal, not all should take full credit. A co-author weighted coefficient is also calculated based on the scheme proposed by Zhang [34] for each author for every co-authored article. This weighted coefficient is a quantitative means to attribute co-authors' credit based on their rank in the author list (See S1.3 in S1 File for the co-author weighted coefficient formula).

This review adopts the T0 to T4 classification system defined by the Institute for Clinical and Translational Research [35] to assess each study's translational research stage. Briefly, T0 denotes basic and preclinical biomedical research; T1 indicates early human studies such as proof of concept or Phase 1 clinical trial; T2 studies involve translation to patients (well-designed Phase 2 and 3 clinical trials); T3 involves translation to practice, which covers dissemination and implementation research; and T4 is for translation to communities. The reviewers assigned a T stage to each included study after data extraction. Any disagreements that arose between the reviewers were resolved through discussion.

## Data analysis and presentation

The search results and the study inclusion process were presented following the guideline of Preferred Reporting Items for Systematic Reviews and Meta-analyses (PRISMA) [36], and specifically, the extension for scoping review (PRISMA-ScR) [37]. See S2 File for the PRISMA-ScR checklist for this report. Extracted data were further tabulated into data tables (See S3 File) using Microsoft Excel 365 (Microsoft Corp, WA, USA). Descriptive and bibliometric analyses were conducted using Microsoft Excel 365 and R Studio (Posit Software, UK), running with R version 4.2.2 [38]. Network analysis was performed using the visNetwork package in R [39] to visualise the complex relationships among the data items.

The reviewers adopted a quantitative approach to analyse the extracted data's bibliometrics, including authors, institutions, publications, references, MeSH terms, and research impacts based on Google Scholar citations to explore the translational status of the field. The potential beneficial actions and positive outcomes of RBAC ingestion on health and the possible applications for any disease conditions were also illustrated in tables and graphs.

## Results

### Sources of evidence

Searches were conducted between September and October 2022. The selection process is depicted in Fig 1. In total, 98 primary research articles were included as sources of evidence, and 13 trial registration records were identified for publication rate assessment. The list of all included articles (n = 98) with their essential characteristics and citation details are provided as supporting information (S4.1 Table in S4 and S5 Files).

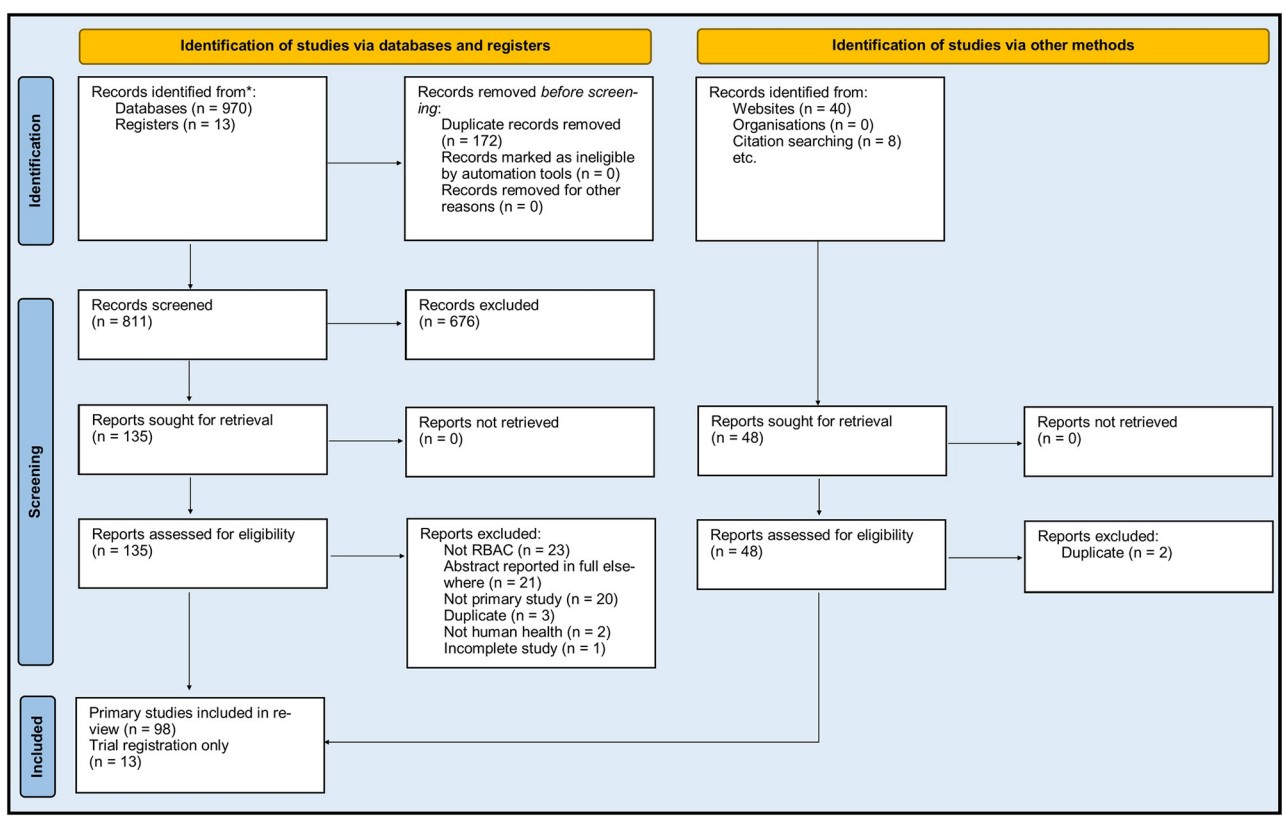

**Fig 1. A flow diagram summarises the selection of evidence sources based on the PRISMA 2020 template.**

Table 2 shows a descriptive summary based on publishing year, country, language, and type. Articles were published between 1998 to 2022 (25 years), with 18.37% (n = 18) considered the more recent publications (2018—2022) and 44.9% (n = 44) published within the last ten years. Although the number of articles published per year ranges from 1 to 12, the publishing rate was linear over time, as illustrated in Fig 2. On average, 3.92±2.8 (mean ± standard deviation) articles were published annually with a median of three per year.

The USA produced the most research output (n = 29, 29.59%), followed closely by Japan (n = 22, 22.45%), South Korea (n = 15, 15.31%), Egypt (n = 11, 11.22%), and Hungary (n = 7, 7.14%). Readers can find a more detailed country-level summary in S4.2 Table in S4 File. Language-wise, 82.65% of the articles (n = 81) were published in English, 11.22% (n = 11) in Japanese, and 6.12% (n = 6) in Korean. This review found no Chinese-language articles fulfilling the eligibility criteria for inclusion. Out of the 98 included articles, 85 (86.73%) are full papers published in peer-reviewed journals. The remaining consists of conference abstracts presenting novel findings not already published elsewhere (n = 8, 8.16%), book chapters (n = 2, 2.04%), and one each of short communication, study protocol, and thesis.

Table 3 summarises the articles based on the study design and translational stage. There are 56 articles (57.14%) reporting preclinical investigation results based on animal models (n = 25, 25.51%), in vitro cell experiments (n = 19, 19.39%), chemical analysis (n = 1, 1.02%), or mixed-method design (n = 9, 9.18%). The remaining 42 articles (42.86%) were human clinical studies consisting of 29 interventional studies (29.59%) and 13 observational studies (13.27%). Most observational studies are case reports (n = 8, 8.16%) or case series (n = 4, 4.08%), except a

**Table 2. Descriptive summary of the included articles: Year, country, language, and type.**

| Summary Characteristics | N (% total) |
|---|---|
| **All included articles** | 98 (100%) |
| *Publication Years* | |
| 2018–2022 | 18 (18.37%) |
| 2013–2017 | 26 (26.53%) |
| 2008–2012 | 18 (18.37%) |
| 2003–2007 | 21 (21.43%) |
| 1998–2002 | 15 (15.31%) |
| *Country* | |
| United States of America (USA) | 29 (29.59%) |
| Japan | 22 (22.45%) |
| South Korea | 15 (15.31%) |
| Egypt | 11 (11.22%) |
| Hungary | 7 (7.14%) |
| Others | 15 (14.29%) |
| *Language* | |
| English | 81 (82.65%) |
| Japanese | 11 (11.22%) |
| Korean | 6 (6.12%) |
| *Publication Type* | |
| Full paper | 85 (86.73%) |
| Abstract | 8 (8.16%) |
| Book Chapter | 2 (2.04%) |
| Short communication | 1 (1.02%) |
| Study protocol | 1 (1.02%) |
| Thesis | 1 (1.02%) |

descriptive cross-sectional study. Randomised controlled trials (RCT) are the most common interventional human clinical studies, with 21 counts (21.43%), followed by single-arm before and after studies (n = 7, 7.14%). There is also a non-randomised controlled interventional trial.

Fig 3 shows the number of articles published over time delineated by study design. There is a clear trend of increasing research attention on human interventional studies in recent years, with 61.11% (n = 11) of articles published between 2018 and 2022 being clinical interventional studies. In comparison, the percentages are 30.77% (n = 8), 16.67% (n = 3), 14.29% (n = 3), and 26.67% (n = 4) in the earlier periods of 2013–2017, 2008–2012, 2003–2007, and 1998–2002, respectively. Based on the T0 to T4 classification system describing where research sits on the translational science spectrum, the preclinical studies (n = 56, 57.14%) were T0 studies. All observational and single-arm interventional (before & after) studies were T1 research. Not all controlled human interventional studies were considered T2 research due to their small sample size ($\leq$ 50). This review identified only four well-designed RCTs as T2 research (n = 4, 4.08%), with the remaining interventional studies belonging to T1 research (n = 38, 38.78%).

Table 4 provides a summary of the sources of RBAC and funding. Although different types of RBAC products are investigated in the included articles, they are linked to three commercial companies, namely Daiwa, Erom, and STR Biotech Co., Ltd. (Chuncheon, South Korea; hereafter referred to as STR Biotech). Biobran MNG-3 from Daiwa was the subject of interventions and investigations in 86.73% of the included studies (n = 85). Eight articles (8.16%) reported

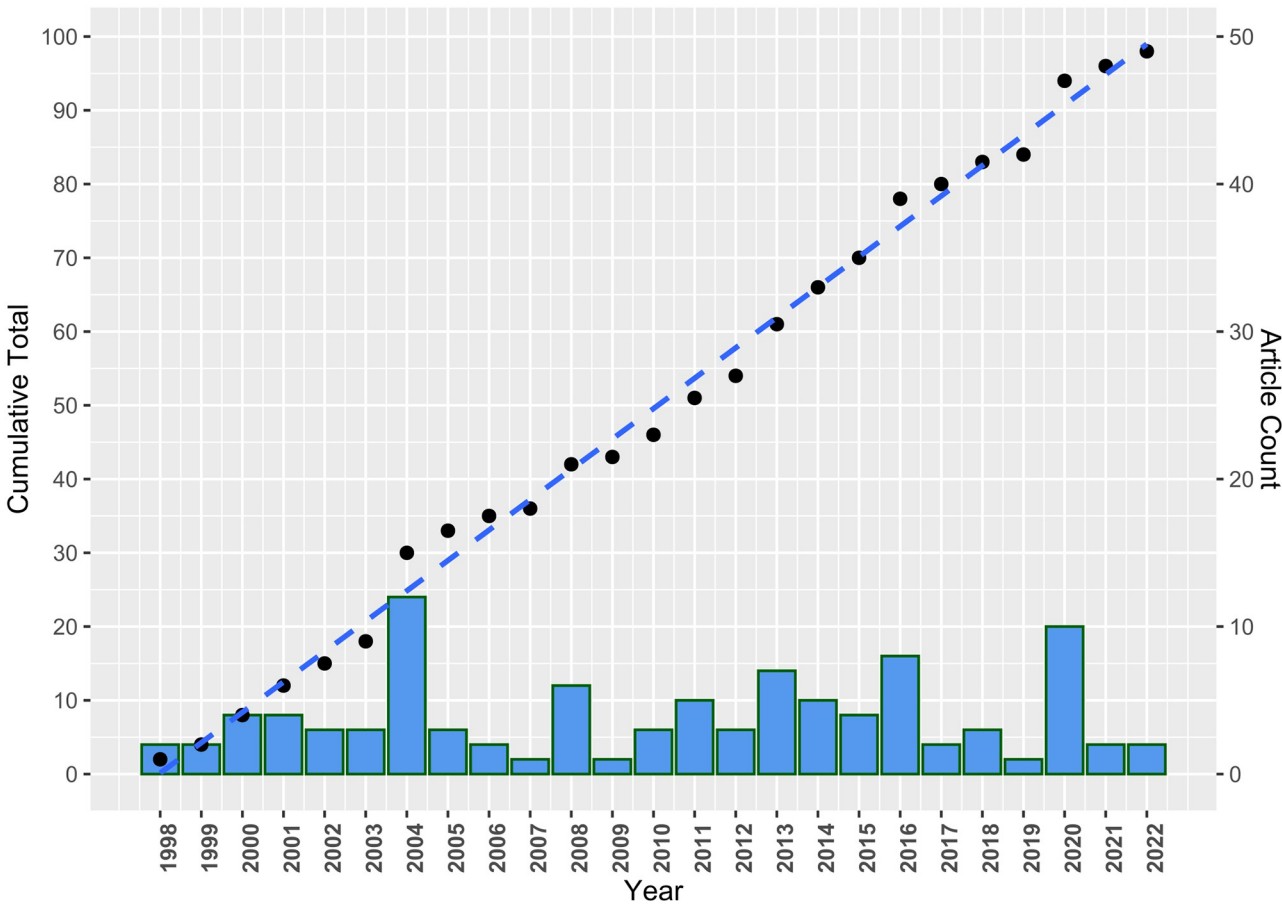

**Fig 2. A scatter plot of the cumulative number of articles published over the years and a bar chart of the annual article count.**

studies on Erom's products, including RBEP, Erom's fermented rice bran, fermented Super-C3GHi bran, and oral nutritional supplement. The remaining five articles (5.1%) were studies based on bioprocessed polysaccharides, fermented black rice bran, or bioprocessed rice bran extract developed by STR Biotech.

Slightly over half of the included articles did not disclose their funding sources (n = 50, 51.02%). Fig 4 shows the proportion of papers that acknowledged their funding sources over time. We observed an increasing disclosure trend, with 72.22% of articles published between 2018 and 2022 making the disclosure, compared to 61.54%, 38.89%, 47.62%, and 13.33% of 2013–2017, 2008–2012, 2003–2007, 1998–2002, respectively. RBAC research received the most financial support from commercial sources, with Daiwa being the highest funder of RBAC research through financially supporting 26 studies (26.53%) and providing products to another eight studies (8.16%). Erom supported five studies (5.10%), and a reseller of Daiwa partially funded two other studies (2.04%). Public funding from government grants or universities is the second most common source, with 17 articles (17.35%) acknowledged as partially or entirely supported by public funds. Private individuals or nonprofit organisations supported six (6.12%) studies financially. For full details on the disclosure statement of all the included articles, please refer to S4.3 Table in S4 File.

Among the 21 articles of controlled trials, three were published before registration was mandated in 2007. Of the remaining 19, 11 were registered, and 8 were not. Thus, the

**Table 3. Descriptive summary of the included articles: Study design and translational stage.**

| Summary Characteristics | | N (% total) |
|---|---|---|
| **All included articles** | | 98 (100%) |
| *Study Design* | | |
| Preclinical | | 56 (57.14%) |
| Animal | 25 (25.51%) | |
| Animal + Cell | 6 (6.12%) | |
| Animal + Cell + Chemical | 3 (3.06%) | |
| Cell | 19 (19.39%) | |
| Cell + Chemical | 2 (2.04%) | |
| Chemical | 1 (1.02%) | |
| Clinical | | 42 (42.86%) |
| *Interventional* | | |
| Randomised controlled trial | 21 (21.43%) | |
| Non-randomised controlled trial | 1 (1.02%) | |
| Before and after study | 7 (7.14%) | |
| *Observational* | | |
| Descriptive cross-sectional study | 1 (1.02%) | |
| Case series | 4 (4.08%) | |
| Case report | 8 (8.16%) | |
| *Translational Stage* | | |
| T0—Basic biomedical research | | 56 (57.14%) |
| T1—Translation to humans | | 38 (38.78%) |
| T2—Translation to patients | | 4 (4.08%) |

registration rate is only 57.89%. The search found 13 trial registration records (See S4.4 Table in S4 File). Among them, seven are listed on ClinicalTrials.gov (USA), two on the University Hospital Medical Information Network Center Clinical Trial Registry (Japan), two on the South Korean Clinical Research Information Service Registry, one on United Kingdom's (UK) Current Controlled Trials, and one on the Australian New Zealand Clinical Trials Registry. Nine trials have been published, with their results found as sources of evidence. Of the four trials with no results published yet, two remain ongoing, one has ended with the publication of results pending, and another has been discontinued due to disruption during the COVID-19 pandemic. Hence, the publication rate of the registered trials is 90%.

## Bibliometric analysis

**Authors.** A total of 289 unique names were extracted from the author lists of the included articles. Table 5 shows the top 8 authors ranked by the sum of their co-author weighted coefficient (TWC). A more detailed summary of the top 10 authors can be found in S4.5 Table in S4 File. The most prolific author in the field is Ghoneum, Mamdooh (TWC = 30.0) of Charles Drew University of Medicine and Science, who is either the first author or a corresponding author of 30 included articles accounted for 30.61% of all primary research in the field. Ghoneum specialises in preclinical studies, particularly in vitro cell-based experiments (53.3% of his published works) and animal models to a lesser extent (30%). Ghoneum was one of the developers of the MGN-3 polysaccharides in 1992 [31]. He first published on RBAC in 1998 and has remained active since. In contrast, another co-developer, Maeda, Hiroaki (TWC = 5.36), was only active between 2000 to 2004.

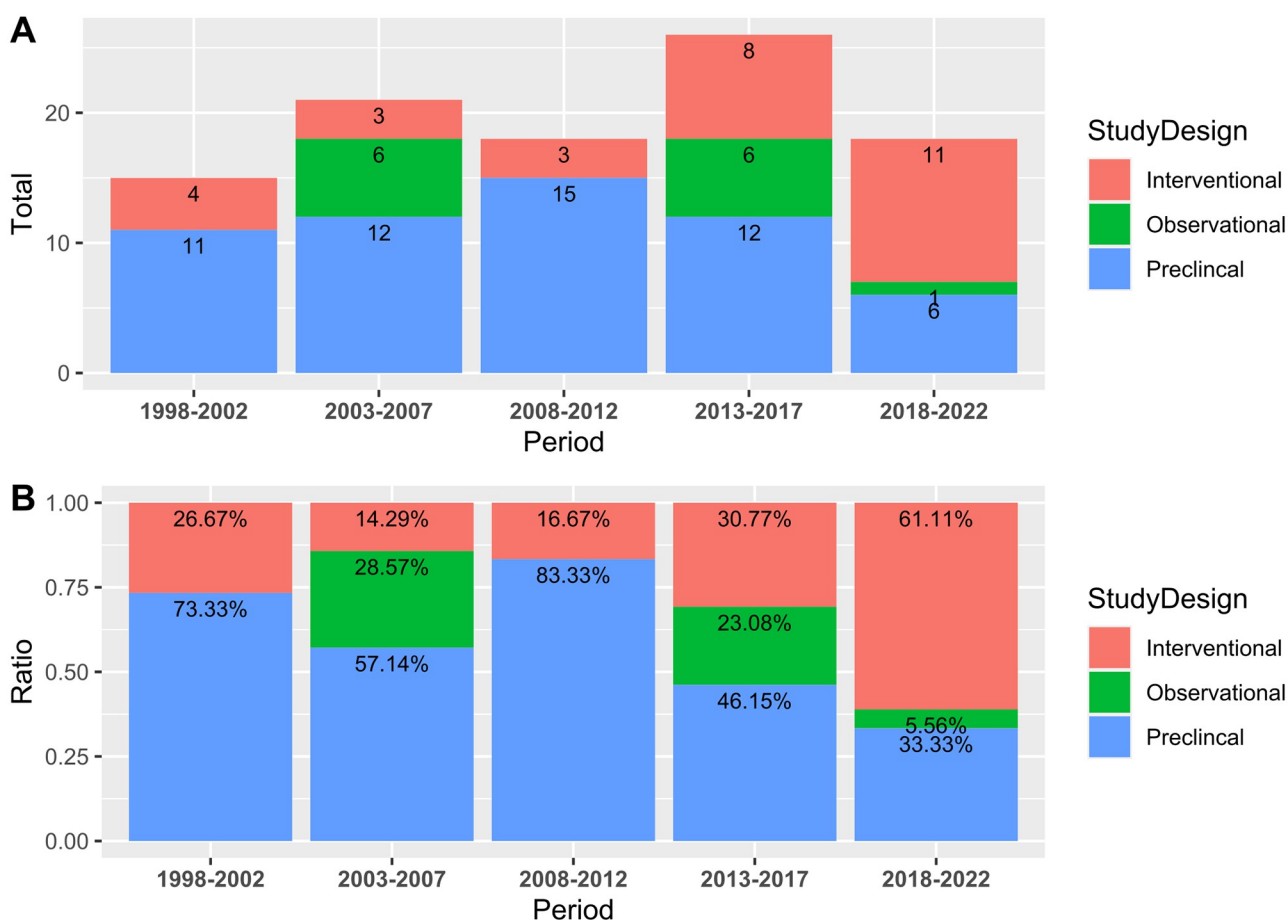

**Fig 3. The number of articles published over time by study design: (A) the absolute count and (B) the relative percentage.**

**Table 4. Descriptive summary of the included articles: Sources of product and fund.**

| Summary Characteristics | | N (% total) |
|---|---|---|
| **All included articles** | | 98 (100%) |
| *Commercial Source of RBAC* | | |
| Daiwa Pharmaceutical Co. Ltd. | | 85 (86.73%) |
| Erom Co. Ltd. | | 8 (8.16%) |
| STR Biotech Co. Ltd. | | 5 (5.10%) |
| *Funding Sources* | | |
| Not Disclosed | | 50 (51.02%) |
| Disclosed | | 48 (48.98%) |
| *Commercial—Daiwa* | 26 (26.53%) | |
| *Public* | 17 (17.35%) | |
| *Commercial—Daiwa (Product Only)* | 8 (8.16%) | |
| *Private / Nonprofit* | 6 (6.12%) | |
| *Commercial—Erom* | 5 (5.10%) | |
| *Commercial—Others* | 2 (2.04%) | |

Note: The sum of the funding source count is greater than the number of articles that disclosed funding sources since a study can have more than one source of funds.

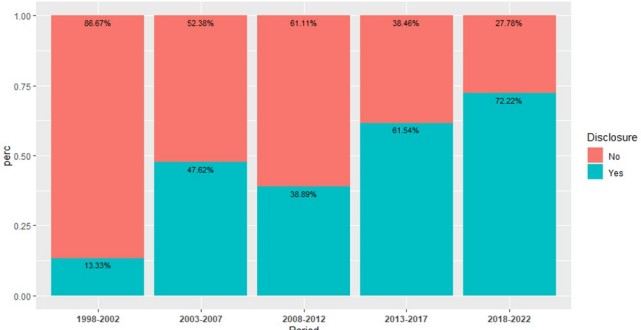

**Fig 4. The proportion of articles published over the years with funding disclosure.**

Other top contributors in order of TWC were Badr El-Din, Nariman K (TWC = 7.64) from the Egyptian University of Mansoura, who specialises in animal experiments of Biobran MGN-3; Gollapudi, Sastry from the University of California at Irvine who collaborated with Ghoneum on in vitro experiments of Biobran MGN-3; Hajtó, Tibor (TWC = 6.0), a Hungarian clinician and academically affiliated with University of Pécs, contributed a series of observational studies based on his clinical experience in the use of Biobran MGN-3; Egashira, Yukari (TWC = 4.5) from Chiba University, Japan was the main driver studying the effects of Biobran MGN-3 on GaIN-induced hepatitis mice models; Lewis, John E (TWC = 4.0) from the University of Miami Miller School of Medicine conducted RCTs with Biobran MGN-3 in several human conditions. Hong, Seong Gil (TWC = 3.75) is the only listed researcher whose works were not focused on Biobran MGN-3. Hong is affiliated with the Erom Research Office and has co-authored articles on Erom's RBAC products since 2005.

The collaborative networks of some of these prominent authors are shown in Fig 5. A complete network diagram is available in S4.1 Fig S4 File and the original interactive network diagram is available online as a reference [40]. The top three most prolific authors (Ghoneum, Badr El-Din, and Gollapudi) formed an international collaboration network between the USA and Egypt for research on Biobran MGN-3. Ghoneum also connected to a group of clinicians in Vietnam via a clinical study. Other collaborative research networks are predominantly at the national level, such as Maeda and Egashira being at the centres of a Japanese network,

**Table 5. The publishing period and study design classification of the top 8 authors.**

| # | Author | Publishing Period | TWC | Article Count (% Total) | Clinical Intervention | Clinical Observation | Preclinical Animal | Preclinical In Vitro | Preclinical Chemical |
|---|--------|-------------------|-----|-------------------------|-----------------------|----------------------|--------------------|----------------------|----------------------|
| 1 | Ghoneum, Mamdooh | 1998–2021 | 30.00 | 30 (30.61%) | 26.7% | - | 30.0% | 53.3% | - |
| 2 | Badr El-Din, Nariman K. | 2008–2020 | 7.64 | 9 (9.18%) | - | - | 88.9% | 11.1% | - |
| 3 | Gollapudi, Sastry | 2003–2011 | 6.00 | 6 (6.12%) | - | - | - | 100% | - |
| 3 | Hajtó, Tibor | 2013–2018 | 6.00 | 6 (6.12%) | - | 100% | - | - | - |
| 5 | Maeda, Hiroaki | 2000–2004 | 5.36 | 7 (7.14%) | 14.3% | - | 57.1% | 14.3% | 28.6% |
| 6 | Egashira, Yukari | 2001–2017 | 4.50 | 6 (6.12%) | - | - | 83.3% | 16.7% | - |
| 7 | Lewis, John E. | 2012–2020 | 4.00 | 4 (4.08%) | 100% | - | - | - | - |
| 8 | Hong, Seong Gil | 2005–2022 | 3.75 | 6 (6.12%) | 16.7% | - | 66.7% | 50.0% | - |

Note: The sum of the article study design percentage can be > 100% since some articles reported studies with multiple types of design. TWC = Sum of weighted coefficient.

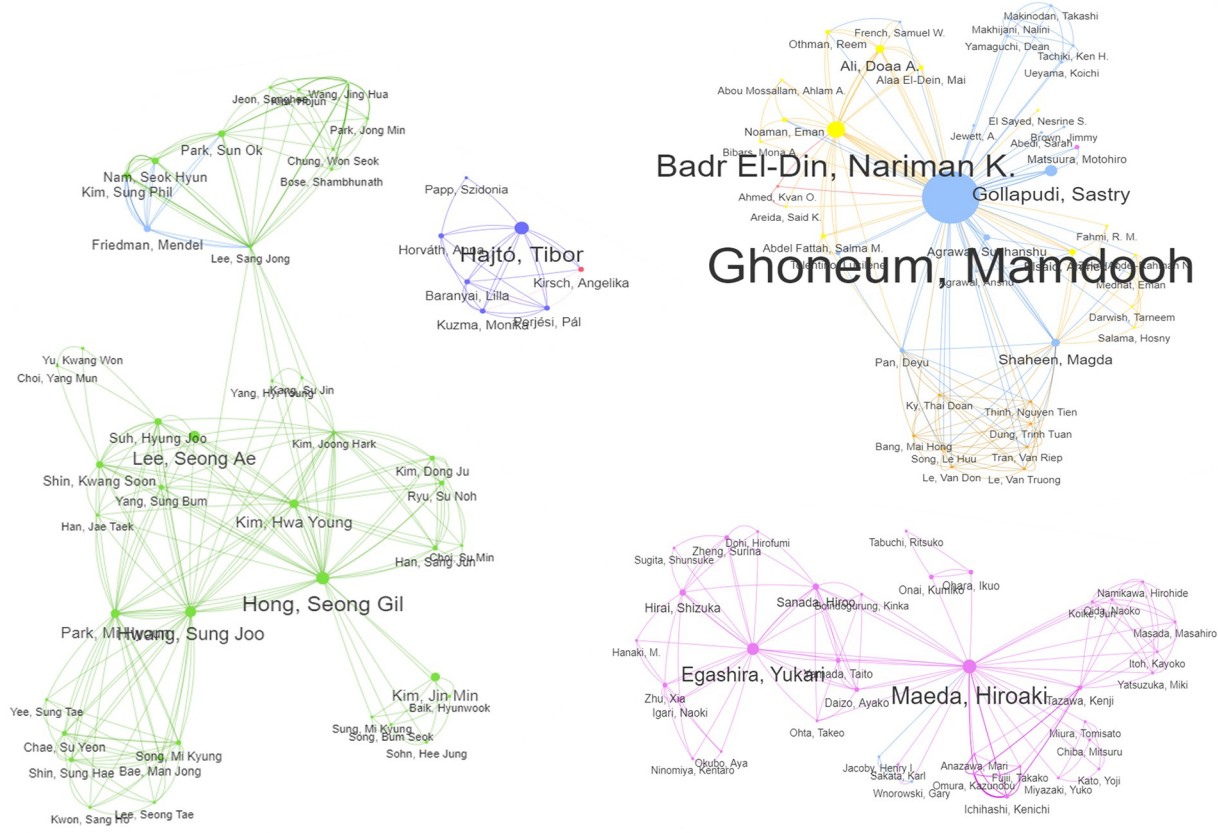

**Fig 5. Collaborative networks of selected key authors.** Each node represents an author. The links between authors represent co-authorships. The size of the nodes and font size of the author's name reflect the co-author weighted coefficients. Author nodes and links are coloured based on their country (Blue = USA; Magenta = Japan; Green = S. Korea; Yellow = Egypt; Purple = Hungary). The original interactive network diagram is available at https://resource.rbac-qol.info.

Hong was linked to an extensive number of Korean authors, Lewis collaborated with a vast network of American clinicians, and Hajtó formed a network with his Hungarian co-authors.

**Institutions.** The 289 authors were affiliated with 100 institutions from 18 countries. Among these institutions, 51 were academic institutions or universities, 17 were healthcare organisations, including hospitals or long-term care centres, 14 were private clinics, 10 were research laboratories, and 8 were commercial entities. Hence, academics, clinicians, and researchers conducted most RBAC research in the public domain. Table 6 listed the top 8 institutions by article count showing the number of affiliated authors and their publishing period. Predictably, the institutions of the top 8 prominent authors were all on this list.

Fig 6 shows the collaborations between some key institutions in network diagrams. The node and font sizes of the names reflect their centrality in the collaborative network. Inevitably, Charles Drew University of Medicine and Science, where Ghoneum is affiliated, is the most prominent institution forming a tight network with the University of Mansoura (Badr El-Din) in Egypt and a small number of other institutions. This network visualisation reveals that even though the primary research was mainly conducted by academic, healthcare, and research institutions, the product companies (Daiwa, Erom and STR Biotech) also have

**Table 6. The top 8 institutions in the field of RBAC.**

| # | Institution | Country | Type | Publishing Period | No. Authors | TWC Authors | Article Count (% Total) |
|---|---|---|---|---|---|---|---|
| 1 | Charles Drew University of Medicine and Science | USA | Academic | 1998—2021 | 6 | 35.03 | 30 (30.61%) |
| 2 | Daiwa Pharmaceutical Co. Ltd. | Japan | Commercial | 2000—2017 | 10 | 10.64 | 11 (11.22%) |
| 3 | University of California at Irvine | USA | Academic | 2003—2021 | 3 | 9.00 | 9 (9.18%) |
| 3 | University of Mansoura | Egypt | Academic | 2008—2020 | 5 | 13.12 | 9 (9.18%) |
| 5 | Erom Co. Ltd. | South Korea | Commercial | 2004—2022 | 13 | 15.49 | 8 (8.16%) |
| 6 | Chiba University | Japan | Academic | 2000—2017 | 12 | 14.49 | 7 (7.14%) |
| 7 | University of Pécs | Hungary | Academic | 2013—2018 | 12 | 14.36 | 7 (7.14%) |
| 8 | University of Miami Miller School of Medicine | USA | Academic | 2013—2022 | 21 | 10.59 | 4 (4.08%) |

Note: The sum of the article study design percentage can be > 100% since some articles reported studies with multiple types of design.

centrality in the research network, albeit to a lesser extent. That is, likely through funding, product sponsorship, or technical assistance. The two South Korean companies, Erom and STR Biotech, notably collaborated in some of their RBAC research.

**Publications.** Seventy unique scientific publications had published articles related to RBAC research, the majority being academic journals (n = 65, 92.86%). The remaining five comprise two edited books, one conference proceeding, one university dissertation repository, and one professional journal. Only 17 of these publications have published at least two primary research articles on RBAC, as shown in Table 7.

Among them, Clinical Pharmacology and Therapy (Yakuri to Rinsho) has the highest article count (Iyaku Shuppan, n = 6, 6.12%), followed by International Journal of Immunopathology and Pharmacology (Sage, n = 5, 5.1%), Anticancer Research (International Institute of Anticancer Research, n = 4, 4.08%), Evidence-based Complementary and Alternative Medicine (Hindawi, n = 3, 3.06%), and Journal of the Korean Society of Food Science and Nutrition

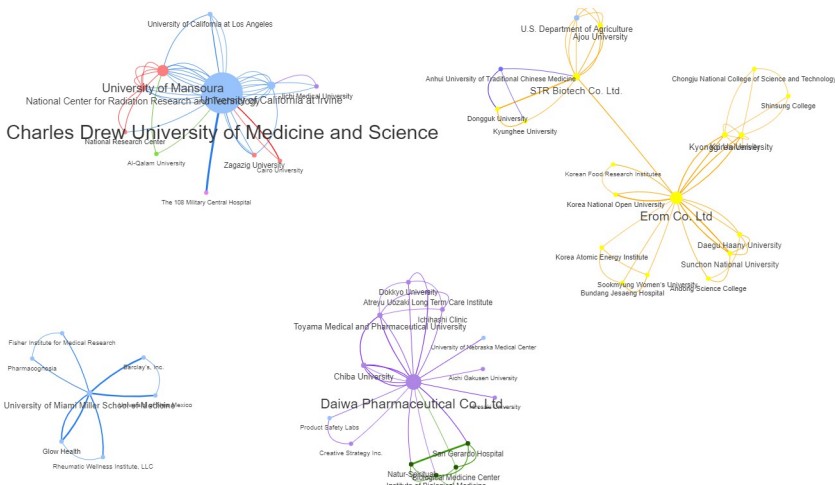

**Fig 6. Collaborative networks of key institutions in the field of RBAC research.** Each node represents an institution. The links between institutions represent co-occurrences. The node and font sizes reflect the centrality measures based on the number of connections. Institution nodes and links are coloured based on their country (Blue = USA; Purple = Japan; Yellow = S. Korea; Red = Egypt; Green = Italy; Magenta = China). The original interactive network diagram is available at https://resource.rbac-qol.info.

**Table 7. The top publications ranked by article count.**

| # | Publications | IF | Cite Score | Publisher | Quartile: Category | Art Count (% Total) | Publishing Period |
|---|---|---|---|---|---|---|---|
| 1 | Clinical Pharmacology and Therapy (Yakuri to Rinsho) | NA | NA | Iyaku Shuppan | NA | 6 (6.12%) | 2004–2004 |
| 2 | International Journal of Immunopathology and Pharmacology | 3.219 | 4.1 | Sage Publications | Q1: Medicine (all) | 5 (5.10%) | 2004–2016 |
| 3 | Anticancer Research | 2.48 | 3.8 | International Institute of Anticancer Research | Q3: Cancer Research; Q3: Oncology | 4 (4.08%) | 2005–2014 |
| 4 | Evidence-based Complementary and Alternative Medicine | 2.629 | 3.0 | Hindawi Publishing | Q1: CAM | 3 (3.06%) | 2014–2020 |
| 4 | Journal of the Korean Society of Food Science and Nutrition | 0.548 | 0.9 | The Korean Society of Food Science and Nutrition | Q3: Food Science; Q4: Nutrition & Dietetic | 3 (3.06%) | 2004–2022 |
| 5 | Biomedicine & Pharmacotherapy | 6.529 | 9.3 | Elsevier | Q1: Pharmacology | 2 (2.04%) | 2020–2020 |
| 5 | Cancer Detection and Prevention (Continued as Cancer Epidemiology from 2009) | 2.984 | 3.9 | Elsevier | Q3: Cancer Research; Q3: Epidemiology; Q3—Oncology | 2 (2.04%) | 2000–2008 |
| 5 | Cancer Letters | 8.679 | 14 | Elsevier | Q1: Cancer Research: Q1: Oncology | 2 (2.04%) | 2003–2008 |
| 5 | Clinical Case Reports and Reviews | NA | NA | Open Access Text | NA | 2 (2.04%) | 2015–2016 |
| 5 | Integrative Cancer Therapies | 3.279 | 4.0 | Sage Publications | Q1: CAM; Q2: Oncology | 2 (2.04%) | 2016–2016 |
| 5 | International Congress on Anti-Aging & Biomedical Technologies | NA | NA | American Academy of Anti-Aging Medicine | NA | 2 (2.04%) | 1999–2000 |
| 5 | Journal of Agricultural and Food Chemistry | 5.279 | 7.3 | ACS Publications | Q1: Agricultural & Biological Sciences (all): Q1: Chemistry (all) | 2 (2.04%) | 2013–2014 |
| 5 | Journal of Dietary Supplements | 2.272 | 3.9 | Informa Healthcare | Q2: Food Science: Q2: Nutrition & Dietetic; Q2: Pharmacology | 2 (2.04%) | 2008–2020 |
| 5 | Journal of Japanese Association for Dietary Fiber Research | NA | NA | Japanese Association for Dietary Fiber Research | NA | 2 (2.04%) | 2001–2002 |
| 5 | Journal of Radiation Research | 2.724 | 3.5 | Oxford Academic | Q2:HTM; Q2: Radiation Q2: Radiology, Nuclear Medicine and Imaging | 2 (2.04%) | 2013–2019 |
| 5 | Neoplasma | 2.757 | 3.3 | AEPress | Q1: Medicine (all); Q3: Oncology | 2 (2.04%) | 2009–2011 |
| 5 | Nutrition and Cancer | 2.9 | 4.1 | Routledge | Q3: Cancer Research; Q2: Medicine (misc); Q2: Nutrition & Dietetic; Q2: Oncology | 2 (2.04%) | 2008–2016 |

Abbreviations: CAM, complementary and alternative medicine; HTM, Health, Toxicology and Mutagenesis; IF, the current impact factor (2-Year)—reported by academic-accelerator.com as of 13/2/2023; NA, not available

(Korean Society of Food Science and Nutrition, n = 3, 3.06%). Cancer Research, Oncology, Nutrition & Dietetics, Medicine, and Complementary & Alternative Medicine are the most common subject categories of these publications.

**Citations.** Of the 98 included articles, 89 (90.81%) have at least one citation in Google Scholar. The mean citation rate is 27.89 ±34.57, with a median of 14.5 (interquartile range = 37.5). Among the included articles, there is also extensive referencing to prior works. Table 8 shows the top sources of evidence with the most citations in Google Scholar and those heavily cited by others in the field. The two most cited papers are the seminal works by Ghoneum, which established Biobran MGN-3 as a novel immunomodulator with therapeutic applications for cancer [11] and HIV [12]. Ghoneum was also the first author of another six papers on the list and a co-author of another two. Only three of the 13 most cited articles in the list are unrelated to Ghoneum. Furthermore, only one of these articles by Kim H. Y., et al. [10] investigated RBAC other than Biobran MGN-3.

**Table 8. Top 10 articles ranked according to google citation count and citations by other included articles.**

| Ranking By: | | Article | Publication | Study Design | Year | Citations: | |
|---|---|---|---|---|---|---|---|
| Google | Others | | | | | Google | Others |
| 1 | 1 | Ghoneum (1998b) | International Journal of Immunotherapy | Before & after | 1998 | 180 | 59 |
| 2 | 2 | Ghoneum (1998a) | Biochemical and Biophysical Research Communications | Cell | 1998 | 146 | 44 |
| 3 | 3 | Ghoneum & Jewett (2000) | Cancer Detection and Prevention | Cell | 2000 | 132 | 35 |
| 4 | 6 | Ghoneum & Matsuura (2004) | International Journal of Immunopathology and Pharmacology | Cell | 2004 | 124 | 25 |
| 5 | 4 | Ghoneum & Abedi (2004) | Journal of Pharmacy and Pharmacology | Animal + Cell | 2004 | 100 | 29 |
| 6 | 9 | Noaman et al. (2008) | Cancer Letters | Animal | 2008 | 98 | 18 |
| 7 | 5 | Ghoneum & Gollapudi (2003) | Cancer Letters | Cell | 2003 | 93 | 27 |
| 8 | - | Kim H.Y. et al. (2007) | Journal of Medicinal Food | Animal | 2007 | 85 | 5 |
| 9 | - | Pérez-Martínez et al. (2015) | Cytotherapy | Animal + Cell | 2015 | 73 | 12 |
| 10 | 9 | Badr El-Din et al. (2008) | Nutrition and Cancer | Animal | 2008 | 69 | 18 |
| - | 6 | Ghoneum & Brown (1999) | Anti-aging Medical Therapeutics | Before & after | 1999 | 61 | 25 |
| - | 9 | Ghoneum & Agrawal (2011) | International Journal of Immunopathology and Pharmacology | Cell | 2011 | 60 | 18 |
| - | 8 | Jacoby et al. (2001) | Journal of Nutraceuticals, Functional & Medical Foods | Animal | 2001 | 41 | 19 |

Fig 7 shows the citation networks of the included articles, with research progressing by building on prior findings. While the earlier works of Ghoneum prominently influenced the subsequent studies, there are a couple of more recent results by other authors that stand out, including Cholujova [41], who studied the effects of Biobran MGN-3 on activating dendritic cells in multiple myeloma patients and Pérez-Martínez [42], who investigated natural killer cell (NKC)-mediated cytotoxicity against neuroblastoma in vitro and in vivo. A separate network of four nodes independent of the main citation tree is also highlighted in Fig 7. These are research works on STR Biotech products not related to Biobran MGN-3.

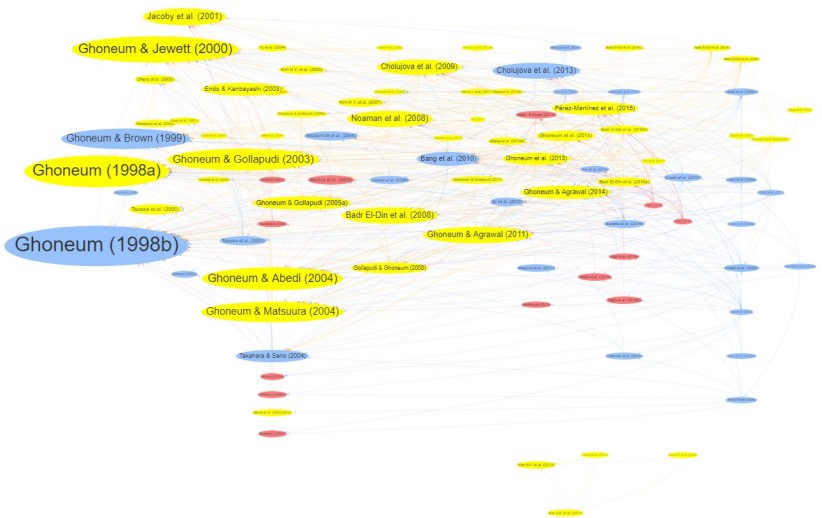

**Fig 7. Citation networks of articles in the field of RBAC research.** Each node represents an article arranged by publishing year from left (earlier) to right (later). The links between nodes are references. The node and font sizes reflect the citation count. The nodes are hierarchically laid based on their years of publication from left to right. The colours denote study types (Yellow = Preclinical, Red = Observational, and Blue = Interventional). The original interactive network diagram is available at https://resource.rbac-qol.info.

**Keywords—MeSH.** Table 9 shows the most common MeSH terms used to characterise the research on RBAC classified under context, method, intervention and outcome. Most research was related to humans (male, female, adults, middle-aged, or aged) or animal models, with neoplasms being the most likely condition for investigation. Method-wise, common keywords include mice, rats (or specifically Wister rats), and organs extracted for analysis in animal experiments, such as liver and spleen. Lipopolysaccharides, or bacteria toxins, are routinely used to induce immunological reactions in preclinical studies of animals or cell lines and measurements done with the enzyme-linked immunosorbent assay.

Many MeSH terms are used to categorise RBAC as a plant-based nutritional intervention, including polysaccharides (polysaccharide MGN3), arabinoxylan, xylans, and hemicellulose. Oryza and shiitake mushrooms are keywords related to RBAC's production process. RBAC is partially water-soluble, with the hydrolysis of RBAC typically used in experiments as a dietary supplement in the included studies. Intraperitoneal injection of RBAC solution was a common practice in animal experiments. Some therapeutic terms that classify RBAC are antineoplastic agents, antioxidants, and immunologic adjuvants or immunologic factors, where actions may possess dose-response relationships.

The outcomes of interest for RBAC research mainly centred on its ability to affect the immune cells, especially NKC, macrophages, and lymphocytes (or T-lymphocytes, in particular). Various cytokines as the signalling proteins of the immune system are also studied, such as interferons (e.g., interferon-gamma), tumour necrosis factor-alpha, and interleukin-6. Upregulation of immune actions on cell proliferation, apoptosis, phagocytosis, and specific gene expressions are outcome measures in many animal and cell-based experiments. Biomarkers such as liver transaminases are investigated in studies as signs of inflammation or oxidative stress. Body weight and quality of life (QoL) measurements are also tracked in many RBAC studies.

Fig 8 offers visualisations of the evolution of research focus in RBAC based on the changing frequency and importance of MeSH terms of outcomes. Early RBAC research published between 1998 to 2002 mainly focused on its effects as an immunologic factor on NKC and T-

**Table 9. Common MeSH terms (occurrence > 5) of the included articles are classified under different categories.** The occurrence count of each keyword is shown in brackets.

| Context | Method | Intervention | Outcome |
|---|---|---|---|
| Humans (59) | Mice (29) | Polysaccharide MGN3 (62) | Killer Cells, Natural (30) |
| Animals (43) | Liver (15) | Arabinoxylan (59) | Cytokines (21) |
| Male (32) | Rats (14) | Xylans (39) | Macrophages (16) |
| Female (27) | Lipopolysaccharides (10) | Oryza (25) | Quality of Life (15) |
| Aged (14) | Spleen (9) | Shiitake Mushrooms (19) | Apoptosis (12) |
| Middle Aged (11) | Enzyme-Linked Immunosorbent Assay (9) | Immunologic Factors (15) | Cell Proliferation (10) |
| Adult (9) | Cell Line (8) | Dietary Supplements (12) | Body Weight (9) |
| Neoplasms (9) | Cell Line, Tumor (8) | Hemicellulose (11) | Inflammation (9) |
| | Rats, Wistar (6) | Polysaccharides (11) | Interferons (9) |
| | | Water (11) | Tumour Necrosis Factor-alpha (9) |
| | | Antineoplastic Agents (10) | Biomarkers (8) |
| | | Injections, Intraperitoneal (9) | Interleukin-6 (7) |
| | | Antioxidants (7) | T-Lymphocytes (7) |
| | | Adjuvants, Immunologic (6) | Transaminases (7) |
| | | Dose-Response Relationship, Drug (6) | Up-Regulation (7) |
| | | Hydrolysis (6) | Gene Expression (6) |
| | | | Interferon-gamma (6) |
| | | | Lymphocytes (6) |
| | | | Oxidative Stress (6) |
| | | | Phagocytosis (6) |

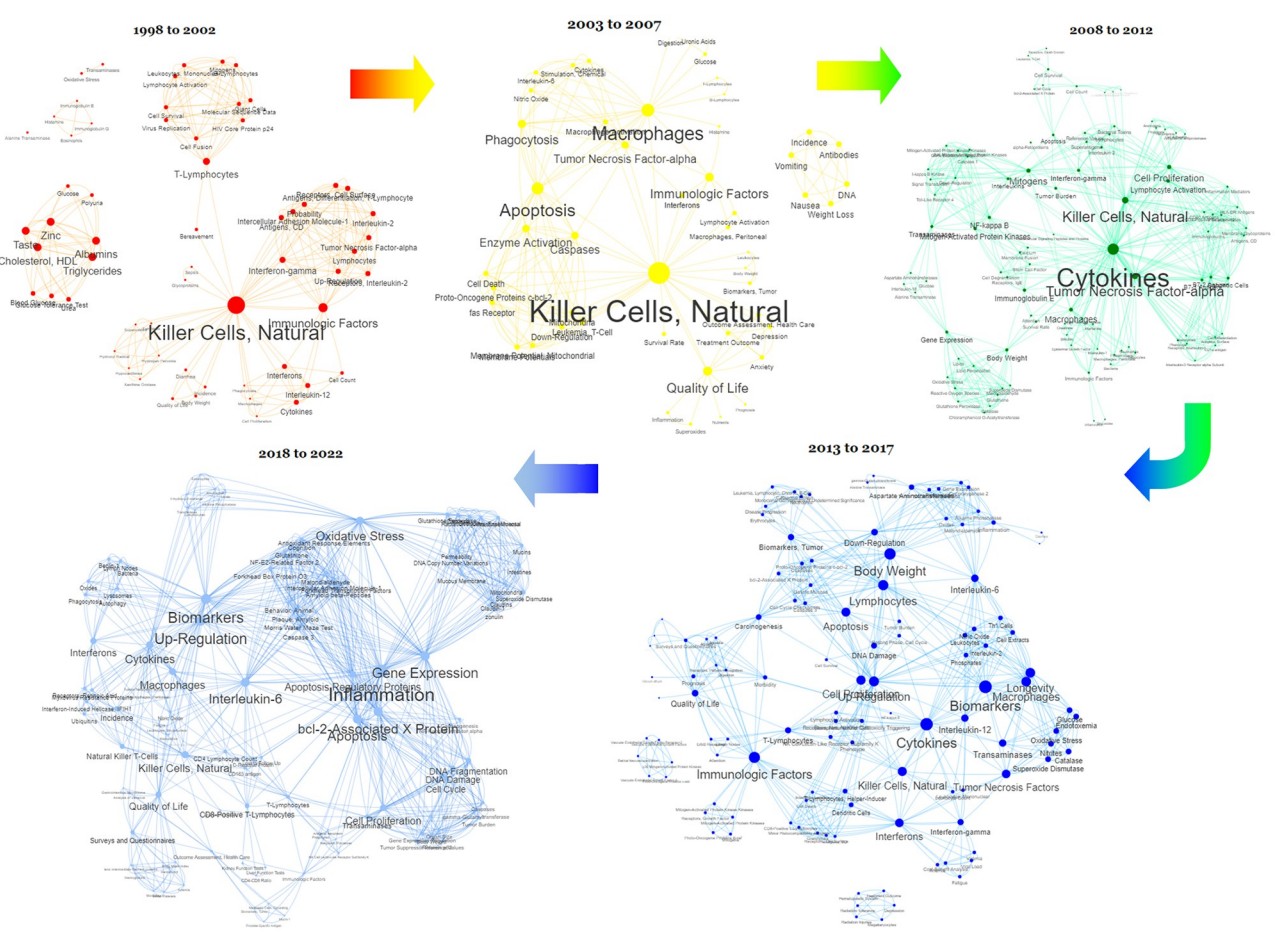

**Fig 8. Networks of outcome keywords (MeSH terms) at different periods reflect the evolution of RBAC research with its shifting focus over time.**
The original interactive network diagram is available at https://resource.rbac-qol.info.

lymphocytes, with some other exploratory research on the effects on lipids and blood glucose. The studies on NKC activation were central in RBAC research from 2003 to 2007. Priming of macrophage phagocytosis and enhancing tumour cell apoptosis were other immunomodulating outcomes featured during this period. We observed a shift in focus with more significant interest in cytokines, including RBAC's effects on tumour necrosis factor-alpha, between 2008 and 2012. More studies during this period were conducted to understand the underlying mechanisms of RBAC beyond NKC. From 2013 onward to 2017, more human observational and interventional studies were conducted with keywords reflecting the systemic immunologic effects of RBAC, such as lymphocytes, cytokines, biomarkers, body weights, and longevity. This trend continued from 2018 to 2022, with more broad-based terms such as inflammation, oxidative stress, biomarkers, gene expression, and up-regulation showing centrality in MeSH networks.

### Research evidence

**Health or disease conditions.** RBAC has been studied for its potential in 17 health or disease conditions, as shown in Table 10. Cancer is the most studied condition, investigated by 45 (45.92%) of the included articles. Among the studies on cancer, 44.44% (n = 20) are preclinical

**Table 10. Health/disease conditions investigated by the included studies ordered by article count and classified by study design.**

| # | Condition | Count (%) | Preclinical (%) | Observational (%) | Interventional (%) |
|---|---|---|---|---|---|
| 1 | Cancer | 45 (45.92%) | 20 (44.44%) | 12 (26.67%) | 13 (28.89%) |
| 2 | Healthy / Nonspecific | 31 (31.63%) | 28 (90.32%) | - | 3 (9.68%) |
| 3 | Hepatitis / Liver Disease | 9 (9.18%) | 7 (77.78%) | - | 2 (22.22%) |
| 4 | Geriatric | 6 (6.12%) | 2 (33.33%) | - | 4 (66.67%) |
| 5 | HIV / AIDS | 4 (4.08%) | 1 (25%) | - | 3 (75%) |
| 6 | Allergy | 4 (4.08%) | 4 (100%) | - | - |
| 7 | CFS | 3 (3.06%) | - | - | 3 (100%) |
| 8 | Gastroenteritis | 3 (3.06%) | 2 (66.67%) | - | 1 (33.33%) |
| 9 | Cold / Flu | 2 (2.04%) | - | - | 2 (100%) |
| 10 | Diabetes mellitus | 2 (2.04%) | 2 (100%) | - | - |
| 11 | Endotoxemia | 2 (2.04%) | 2 (100%) | - | - |
| 12 | Chemical exposure | 1 (1.02%) | - | - | 1 (100%) |
| 13 | IBS | 1 (1.02%) | - | - | 1 (100%) |
| 14 | Rheumatism | 1 (1.02%) | - | 1 (100%) | - |
| 15 | Alzheimer's disease | 1 (1.02%) | 1 (100%) | - | - |
| 16 | Bacterial infection | 1 (1.02%) | 1 (100%) | - | - |
| 17 | Oxidative stress | 1 (1.02%) | 1 (100%) | - | - |

Abbreviation: AIDS, acquired immunodeficiency syndrome; CFS, chronic fatigue syndrome; HIV, human immunodeficiency virus; IBS, irritable bowel syndrome.

**Note:** The sum of all article counts is > 100%, as some articles reported results related to more than one condition.

experiments, 26.67% (n = 12) are observational, and 28.89% (n = 13) are clinical interventional studies. The most common cancer sites investigated are breast (n = 8), liver (n = 5), and blood (n = 4), although some studies reported results from patients with various cancer sites (n = 9). RBAC has also been studied in colorectal, lung, ovarian, stomach, skin, cervical, head & neck, bile duct, pancreatic, umbilical, uterus and prostate cancers. For more details on the distributions of the study design for each cancer site, please see S4.6 Table in S4 File.

Another 31 articles reported the study outcomes of RBAC in healthy samples. Up to 90.32% of these studies are preclinical experiments (n = 28) on cell lines or animals with no specific diseases or conditions, whereas the remaining three (9.68%) are clinical trials conducted among healthy adults. Liver diseases, including hepatitis, are the third most studied condition, the focus of 9 articles representing 9.18% of all available research, with 77.78% of these studies (n = 7) investigating RBAC's effect in animal models of acute liver injuries. There are also two (22.22%) RBAC clinical interventional studies on liver diseases, one with non-alcoholic fatty liver disease and another with hepatitis C infection.

The potential use of RBAC for geriatric disease prevention among the older population was the subject of another two preclinical studies and four clinical trials (n = 6 total, 6.12% of all research), making it fourth on the list. Other health or disease conditions that have been studied in humans with RBAC as an intervention include HIV (n = 3), chronic fatigue syndrome (CFS, n = 3), gastroenteritis (n = 1), cold/flu (n = 2), chemical exposure (n = 1) and irritable bowel syndrome (IBS, n = 1). The effect of RBAC against rheumatism was reported in one case series. The use of RBAC for allergy, diabetes mellitus, endotoxemia, Alzheimer's disease, bacterial infection, and oxidative stress are conditions studied in preclinical research but have yet to be translated to humans.

**Beneficial actions.** Immunomodulation is the most investigated beneficial action of RBAC, as shown in Table 11. The immune modifying effect of RBAC was the subject in

**Table 11. Reported beneficial actions of RBAC ordered by article count and classified by study design.**

| # | Beneficial Actions | Count (%) | Preclinical (%) | Observational (%) | Interventional (%) |
|---|---|---|---|---|---|
| 1 | Immunomodulation | 36 (36.73%) | 21 (58.33%) | 1 (2.78%) | 14 (38.89%) |
| 2 | Synergistic anticancer effect | 19 (19.39%) | 7 (36.84%) | 10 (52.63%) | 2 (10.53%) |
| 3 | Hepatoprotection | 13 (13.27%) | 10 (76.92%) | 1 (7.69%) | 2 (15.38%) |
| 4 | Anticancer | 10 (10.20%) | 9 (90%) | 1 (10%) | - |
| 5 | Psychoneuroimmuno-modulation | 8 (8.16%) | - | 2 (25%) | 6 (75%) |
| 6 | Antiinflammation | 8 (8.16%) | 7 (87.50%) | 1 (12.5%) | - |
| 7 | Antioxidant | 7 (7.14%) | 7 (100%) | - | - |
| 8 | Radioprotection | 3 (3.06%) | 2 (66.67%) | - | 1 (33.33%) |
| 9 | Chemoprevention | 3 (3.06%) | 3 (100%) | - | - |
| 10 | Antiallergy | 3 (3.06%) | 3 (100%) | - | - |
| 11 | Antibacterial | 3 (3.06%) | 3 (100%) | - | - |
| 12 | Antifatigue | 2 (2.04%) | - | - | 2 (100%) |
| 13 | Antiflu | 2 (2.04%) | - | - | 2 (100%) |
| 14 | No significant effect | 2 (2.04%) | - | - | 2 (100%) |
| 15 | Gastroprotection | 2 (2.04%) | 1 (50%) | - | 1 (50%) |
| 16 | Antihyperlipidemic effect | 2 (2.04%) | 2 (100%) | - | - |

Other benefits that have one count each:

Antiangiogenesis; Antiasthma; Antihyperglycemic effect; Antimetastatic effect; Antiretroviral; Antiviral; Antirheumatic effect; Antiwasting; Chemoprotection; Endothelial improvement; Memory enhancer; Noncytotoxic, Taste influencer.

Abbreviation: QoL, quality of life.

36.73% (n = 36) of all research covering preclinical (n = 21), observational (n = 1) and human interventional (n = 14) studies.

RBAC was also reported to have anticancer actions through activating immune cells against malignant tumours and worked synergistically with other anticancer agents such as chemotherapy drugs and natural substances such as curcumin, baker's yeast and lectin mistletoe extract. The anticancer and synergistic anticancer actions of RBAC were supported by evidence from 29 studies (29.59% of research). Among them were 16 preclinical studies, 11 observational studies, and two interventional studies. RBAC was also shown to have hepatoprotective action based on results from ten preclinical studies, one observational study and two clinical trials. The immunologic effects of RBAC also appeared to have positive psychoneurological impacts on the patients as investigated in eight human studies (8.16% of all research): two observational and six interventional.

Other well-documented benefits of RBAC supported by human studies include radioprotection, antifatigue, antiflu, and gastroprotection. Not all human studies reported positive results. RBAC was found to have no significant effects in a clinical trial for HIV patients and another for CFS. There is also preclinical evidence supporting RBAC to have antiinflammation, antioxidant, chemoprevention, antiallergy, antibacterial, and antihyperlipidemic effects.

**Positive outcome measures.** Table 12 shows the frequently reported positive outcome measures, which closely resemble the common outcome MeSH terms used to characterise the research. Positive outcome measures of RBAC's impact on the immune system most often involve modulating cytokines (n = 25, 25.51%), upregulating NKC (n = 19, 19.39%), activating phagocytosis of macrophages (n = 13, 13.27%), and affecting lymphocytes (n = 9, 9/18%), primarily through inducing proliferation of T & B lymphocytes (n = 11, 11.22%).

**Table 12. Reported positive outcome measures of RBAC ordered by article count and classified by study design.**

| # | Positive Outcomes | Count (%) | Preclinical (%) | Observational (%) | Interventional (%) |
|---|---|---|---|---|---|
| 1 | Cytokines | 25 (25.51%) | 19 (76%) | - | 6 (24%) |
| 2 | Cancer Proliferation | 21 (21.43%) | 14 (66.67%) | 7 (33.33%) | - |
| 3 | Safety & Adverse Events | 19 (19.39%) | 4 (21.05%) | 1 (5.26%) | 14 (73.68%) |
| 4 | Natural Killer Cells | 19 (19.39%) | 9 (47.37%) | - | 10 (52.63%) |
| 5 | Treatment Response | 19 (19.39%) | 1 (5.26%) | 9 (47.37%) | 9 (47.37%) |
| 6 | Survival Rate | 19 (19.39%) | 9 (47.37%) | 8 (42.11%) | 2 (10.53%) |
| 7 | Liver Function Markers | 18 (18.37%) | 12 (66.67%) | 2 (11.11%) | 4 (22.22%) |
| 8 | QoL Assessment | 15 (15.31%) | - | 8 (53.33%) | 7 (46.67%) |
| 9 | Inflammatory Markers | 13 (13.27%) | 10 (76.92%) | 1 (7.69%) | 2 (15.38%) |
| 10 | Macrophages | 13 (13.27%) | 13 (100%) | - | - |
| 11 | T & B Cells Proliferation | 11 (11.22%) | 8 (72.73%) | - | 3 (27.27%) |
| 12 | Gene Expression | 11 (11.22%) | 10 (90.91%) | - | 1 (9.09%) |
| 13 | Apoptosis | 11 (11.22%) | 11 (100%) | - | - |
| 14 | Lymphocytes | 9 ((9.18%) | 5 (55.56%) | - | 4 (44.44%) |
| 15 | Tumour Markers | 8 (8.16%) | - | 6 (75%) | 2 (25%) |
| 16 | Chemo Side Effects | 6 (6.12%) | 2 (33.33%) | 1 (16.67%) | 3 (50%) |
| 17 | Oxidative Stress Markers | 6 (6.12%) | 6 (100%) | - | - |

Other positive outcome measures:

Incidence Rate (5), Dendritic Cells (5), Nitric Oxide Production (5), Mast Cells (3), Neutrophils (2), Histamine (2), Immunoglobulins (2), Eosinophils (1), Hematopoietic Tissues (1).

Abbreviation: QoL, quality of life

Positive outcomes of RBAC reported from cancer-related studies also include reducing cancer proliferation (n = 21, 21.43%) by regulating gene expression (n = 11, 11.22%) and apoptosis (n = 11, 11.22%), increasing the chance of survival (n = 19, 19.39%), enhancing treatment response (n = 19, 19.39%), improving QoL (n = 15, 15.31%), normalising tumour markers (n = 8, 8.16%) and lessening oncological treatment side effects (n = 6, 6.12%). RBAC also appears to favourably affect several biomarkers, including liver function (n = 18, 18.37%), inflammatory (n = 13, 13.27%), and oxidative stress (n = 6, 6.12%). Notably, RBAC was reported to be safe and not associated with any major adverse events (n = 19, 19.39%).

**Visualisation of evidence.** The research evidence on the potential beneficial effects of RBAC against cancer with their associated positive outcome measures can be visualised in Fig 9. The illustration shows that RBAC's immunomodulation action and the related neuro-psychological effect are the most beneficial to cancer patients, with supporting evidence mainly from human interventional studies.

Similarly, the research evidence on the potential health-promoting effects of RBAC for disease prevention in healthy and aged populations with their associated positive outcome measures can be visualised in a network diagram, as shown in Fig 10. Again, the primary action of interest is RBAC's immunomodulation effects supported by mostly preclinical data. Additional visualisation of the effects and outcomes for hepatitis/liver diseases is available as S4.2 Fig in S4 File.

## Discussion

RBAC research progressed steadily with linear growth over a quarter of the century, dominated by works on Biobran MGN-3 through Ghoneum and his collaborative network of

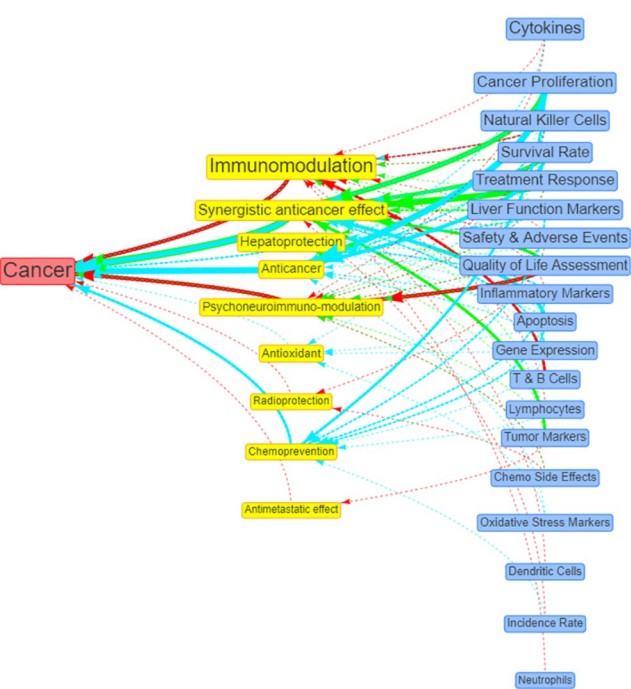

**Fig 9. Network visualisation of the beneficial actions (yellow nodes) of RBAC against cancer (red node) and its associated positive outcomes (blue).** The links between nodes represent the availability of sources of evidence, with the colours indicating the study types (red = interventional, green = observational, and blue = preclinical). The link thickness and node size reflect the number of sources. The original interactive network diagram is available at https:// resource.rbac-qol.info.

international researchers, mainly from Charles Drew University of Medicine and Science (USA), University of California at Irvine (USA), and University of Mansoura (Egypt). Ghoneum continues to be active in RBAC research, with his latest paper published in early 2023 during the preparation of this manuscript (hence not included in this review), demonstrating the antiviral activities of Biobran MGN-3, in vitro and in silico, against the Severe Acute Respiratory Syndrome Coronavirus 2 (SARS-CoV-2) for potential application in COVID-19 prevention and treatment [43]. Nonetheless, the field also received contributions from academics, clinicians, and researchers from 18 countries covering North America, Europe, Asia, and Australia. Other notable authors include Badr El-Din from the University of Mansoura (Egypt), Gollapudi from the University of California at Irvine (USA), Hajtó from the University of Pécs (Hungary), Maeda of Daiwa (japan), Egashira from Chiba University (Japan), Hong from Erom (South Korea), and Lewis from the University of Miami Miller School of Medicine (USA). Most of these authors, except Maeda and Gollapudi, continued active in RBAC research. Hence, individual researchers' interest in the topic appeared to be the main driver of the field. For instance, Hajtó et al. [44] continued to study RBAC's effects on type 1 innate immune system in a diabetic-rat model with a new article published after the completion of the search in this review. Hence, continual monitoring and periodic updates on this scoping review will be needed to track the ongoing research progress.

Our findings showed that the top 3 countries that produced the most research on RBAC are the USA, Japan, and South Korea, with more than 2/3 (67.35%) of all papers originating from these countries. Although the USA, Japan, and South Korea are rice-producing countries, they were not top producers on the global stage. According to the World Rice Production 2022/

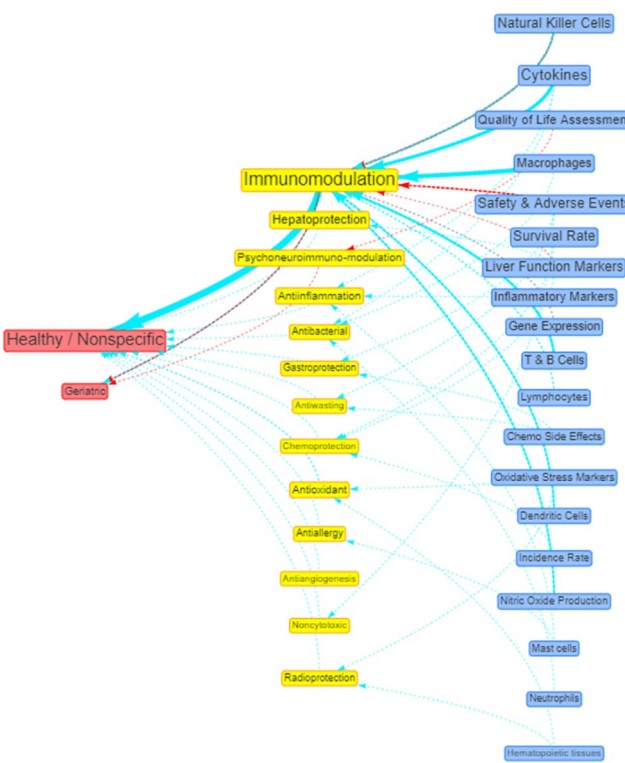

**Fig 10. Network visualisation of the beneficial actions (yellow nodes) of RBAC in healthy or geriatric subjects (red node) and its associated positive outcomes (blue).** The links between nodes represent the availability of sources of evidence, with the colours indicating the study types (red = interventional, green = observational, and blue = preclinical). The link thickness and node sizes reflect the number of sources. The original interactive network diagram is available at https://resource.rbac-qol.info.

2023 estimated by the US Department of Agriculture, Japan, USA, and South Korea, produced 7.45, 5.218, and 3.763 million metric tons of rice and ranked 9th, 13th, 15th of all 83 rice-producing nations in volume, respectively [45]. Compared to the top two global producers of China (147 million metric tons) and India (124 million metric tons), rice production in these countries pales in comparison [45]. However, Japan, the USA, and South Korea are all first-world economies and high-income nations [46]. Thus, research for better economic use of rice bran and derivatives is more prevalent in developed countries than developing nations, irrespective of their global rice production. Moreover, the strong interest in RBAC research in these countries could also be linked to the growing nutraceuticals and functional food market domestically and globally, totalling USD396 Billion in 2021, with North America accounting for 38.5% of the global market values [47]. Hence, economic interests should be the main driving force behind RBAC research.

One surprising outcome of this scoping review is that only one RBAC study published in English originated from China [48]. While we expected some parallel research in Chinese language literature at the onset of the study, the results revealed otherwise. With China already overtaking the USA with the highest research outputs on the Nature Index [49], we have not seen more RBAC research produced in China. Nevertheless, as China is the world's largest rice producer [45], we did find research on other arabinoxylan extracts from rice bran. For example, one study explored the potential health effects of a rice bran arabinoxylan extract with 83% purity on obesity and metabolic inflammation [50]. Other Chinese studies also investigate the

enzymatical processing of rice bran with *Hericium erinaceum* [51], *Inonotus obliquus* [52], *Grifola frondosa* [53], and other basidiomycete fungi [54] to enhance the antioxidant and immune activity of the polysaccharide extracts. However, these studies were excluded from the current review as the bioconversion process did not involve using *L. edodes* enzyme. Thus, other enzymatically treated rice arabinoxylan extracts were investigated by Chinese researchers to enhance the value of rice bran beyond agricultural waste [53]. However, most research from China was conducted in academic institutions and had no follow-up studies beyond the basic experiments. This could be due to the lack of sustainable financial support for translational research [55].

Our analysis reveals that Daiwa, the commercial company producing Biobran MGN-3, also played a central role in advancing research in this field, besides being the top funding provider. Although RBAC products were also developed by two other companies (Erom and STR Biotech), human translational research has been conducted almost solely with Biobran MGN-3. Such an observation further confirmed that RBAC research is primarily commercially driven. As identified by Fabbri et al. [56], in the medically-related industry, commercial sponsorship tends to prioritise research on interventions involving products with a focus on high-income markets. The trend of RBAC research over the years is consistent with such a narrative, with bulks of the research focus on the physiological activities of RBAC that could potentially be marketed as health benefits to target the lucrative nutraceuticals market. Inevitably, industry agendas drove the research agenda away from the more fundamental focus on public health and environmental sustainability issues [57].

The list of top publications that publish RBAC research includes Clinical Pharmacology and Therapy, International Journal of Immunopathology and Pharmacology, Anticancer Research, and Evidence-based Complementary and Alternative Medicine. The list reflects the focus on applied research in this field. Based on the high citation counts of some articles and their appearances in high-impact scholarly journals, RBAC research has influenced various fields of study, including Cancer Research, Oncology, Nutrition & Dietetics, Medicine, and Complementary & Alternative Medicine. For example, RBAC research has been featured in research characterising as a functional ingredient of defatted rice bran [58] and a fermented rice bran by-product with anticancer properties [22, 59–61], as a polyphenolic compound based on hemicelluloses [62] or polysaccharides [63] beneficial for gastrointestinal disorders and cancer [64], and as natural food to protect against gamma-induced oxidative damage [65]. RBAC is also commonly cited as a representative arabinoxylan in the diverse group of dietary fibres that exist in food [66] with anticancer and antioxidant effects [67] and the capacity to modulate gut microbiota [68] for combating human chronic diseases [69]. Results from RBAC research also affected the research design of oncological experimental models [70], especially in preventing tumour growth [71]. Therefore, RBAC has exerted broader scientific contributions to related fields beyond its narrow intended application as a nutraceutical.

The main beneficial effects of RBAC on health and diseases are immunomodulation, synergistic anticancer, and hepatoprotection based on the aggregated summary of evidence count and keyword analysis found in this scoping review (Table 11). The findings are consistent with the reviews of other authors [14, 20, 22, 23, 72]. RBAC was shown to be an immunomodulator by activating NKC [10, 11, 41, 42, 73–79] and macrophages [9, 42, 80–88]. Notably, the ability to enhance NKC cytotoxic activity has also been discussed in many other reviews [14, 20, 22, 89]. RBAC also exerted influences over other immune cells, such as T and B lymphocytes [9, 75, 76, 78, 80, 82, 90–92], dendritic cells [41, 86, 90, 91, 93] and neutrophils [94, 95], by affecting the production of cytokines to regulate their activities [78, 82–84, 94, 96, 97]. RBAC is not cytotoxic to healthy cells [98] but possesses a synergistic anticancer effect that works well with other anticancer agents [99–117] to increase cancer cell apoptosis [99, 100, 102–105, 118–120],

reduce cancer proliferation [10, 99, 100, 102, 107–112, 118, 120–127], improve treatment response [79, 101, 107–112, 114, 116, 128], and enhance chances of survival [10, 76, 101, 102, 106, 109–113, 115, 117, 126, 127, 129]. Hence, an earlier review highlighted that RBAC could be an adjuvant to cancer treatment, with RBAC working synergistically with IL-2, NKC, and chemotherapeutic agents to overcome the dysregulated apoptosis associated with excess oxidative stress in malignancy [21].

RBAC could also benefit patients through the psycho-neuro-immune axis, with cancer patients reporting reduced side effects from chemoradiotherapy and improving QoL [79, 97, 116, 117, 128, 130, 131]. A possible pathway that RBAC could improve the QoL is through improving behaviour comorbidities via reducing systemic inflammation and nutritional status of cancer patients [132]. RBAC also appeared to possess hepatoprotective effects through observed outcomes such as normalising liver transaminases, cytokine regulation, and suppressing the inflammatory signalling pathways [48, 85, 115, 118, 122, 133–139]. A recent review by Egashira [17] also concurred that RBAC prevented liver damage in hepatitis by inhibiting the NK-κB and JNK/MAPK expression. Other notable beneficial actions of RBAC include antiinflammation [48, 90, 96, 126, 139–142], antioxidant [85, 98, 120, 142–145], radioprotection [116, 142, 143], chemoprevention [118, 121, 122], antiallergy [96, 98, 123, 126, 146], antibacterial [84, 86, 94], antifatigue [131, 147], antiflu [147–149], gastroprotection [129, 147–150], and antihyperlipidemia [151, 152].

RBAC has also been studied for potential applications in 15 other conditions apart from healthy subjects and cancer patients. These conditions include liver diseases [48, 85, 133–139], HIV [12, 153–155], CFS [131, 147, 156], IBS [157], common cold/flu [148, 149], Alzheimer's [144], and diabetes [151, 152]. Endo et al. [15] pointed out that many lifestyle-related chronic diseases were caused by oxidative stress leading to chronic inflammation through the "friendly fire" of immune dysregulation. RBAC could alleviate this. Another review by Jason [158] revealed that arabinoxylans in rice bran and arabinoxylan-derived compounds such as ferulic acid and feruloylated oligosaccharides could be the principal antiinflammatory agents acting through transcription factor regulation, gene transcription, enzyme activity, and inflammatory mediator secretions. However, the research on the compositions of RBAC remained limited, with only one study to date attempting to isolate and confirm the active ingredients of RBAC to be arabinoxylan-rich polysaccharides [87].

Regarding translational status, only four well-designed RCTs were classified as T2 research involving translation to patients [79, 101, 116, 148]. The progress has been slow for 25 years of research. However, it is not unexpected as RBAC, or Biobran MGN-3 in particular, is not intended to be a pharmaceutical product but as a nutraceutical or dietary supplement. While pharmaceutical products are regulated to justify their therapeutic efficacies supported by evidence from well-designed Phase III clinical trials before approval, dietary supplements are not subjected to such requirements as long as no therapeutic claims are made. Hence, research on a dietary supplement's therapeutic effect and efficacy is a largely post-market initiative, if any, rather than a prerequisite to market [159]. Nevertheless, there is evidence of growing translational efforts (in the number of human interventional studies) to demonstrate the efficacy of Biobran MGN-3 as an immunomodulator for various groups of patients.

This review found a low registration rate of interventional RBAC trials, with only slightly more than half (57.89%) of the published controlled clinical trials pre-registered on public registries. Again, the low registration rate can be due to RBAC being a nutraceutical or functional food and thus not treated as a therapeutic product, especially for studies among healthy participants or pilots with small sample sizes. Among the registered trials, though, the publication rate was high at 90%, with only one disrupted due to COVID-19 [160]. Furthermore, two reported negative results in the literature showing interest in negative research or findings of

RBAC [153, 156]. However, with the low registration rate of interventional RBAC trials, the risk of publication bias or selective reporting cannot be ruled out, which could threaten the validity of available evidence.

The strength of this scoping review is the use of quantitative bibliometric analysis to characterise the translational progress of a field based on the data extracted from systematic searches and reviews of the literature. Using network analysis to visualise the research field also facilitated the identification of complex patterns and relationships not previously understood. For instance, the institution network diagram (Fig 6) clearly shows commercial companies that produced RBAC products also played a crucial role in enabling the research. All three product companies took centrality in their respective network of collaborative institutions. The visualisation and comparison of RBAC research over different periods (Fig 8) was also a novelty. This review adopted the research trend visualisation with MeSH term similar to the method of Yang and Lee [161]. Through mapping the MeSH terms, this review is the first to identify the core research focus of each period and the growing understanding of RBAC's immune-modulating capabilities over time from a narrow NKC focus [11, 76] to multi-prone systematic effects [41, 42, 142, 162]. Overall, the shifting focus reflects an increasing maturity of the field over 25 years, from benchtop discoveries based on cellular activities to a broad range of translational research in human applications consistent with the observations by other authors [14, 72, 163]. In addition, the condition-benefit-outcome mapping networks (Figs 9 and 10) show how RBAC could potentially benefit cancer patients and healthy populations, which answers the research question of this review visually and succinctly.

This review is not without limitations. With the sizable number of included sources of evidence, it is impossible to present each article's salient point individually. The attempt to aggregate the biological effects and interventional outcomes into high-level keywords such as immunomodulation, hepatoprotection, antiinflammation, QoL, liver function markers etc., overlooks the minor dissimilarities across study contexts, methodology, and findings. As the devil is in the details, this scoping review presents a broad summary of the translational research of RBAC and its potential beneficial effects on health and disease conditions. Further analysis should be conducted to delve into the evidence to elaborate on the detailed composition of RBAC and the possible mechanisms of actions that drive the physiological effects.

This review lacks critical appraisals and quality assessments of the sources of evidence. A previous evidence-based review assessed that unclear risks of bias existed in at least one or more items in selection bias, performance bias, and detection bias of the RCTs on RBAC as a complementary cancer therapy [19]. The current study did not perform any quality assessment of the included studies for two reasons. Firstly, most quality assessment tools, like those proposed by the US National Heart, Lung, and Blood Institute, apply only to human studies [164]. As this scoping review covers all in vitro, animal, and human studies, there is no one set of quality assessment tools suitable for such a wide range of studies. Additionally, the Grading of Recommendations Assessment, Development, and Evaluation (GRADE) system recommended that rating of quality of evidence is best conducted for each specific outcome in a focused system review and meta-analysis and not at the study/trial level as a whole [165]. The current scoping review has a broad research question for mapping a research field with bibliometric analysis in focus; hence, assessing the quality of included study is deferred. Quality assessment should be done in further investigation with a more specific clinical question in the patients, intervention, comparator, and outcomes (PICO) format. Future research should also critically examine the methodology and results of each included article to clarify the mechanistic aspects of RBAC and how it exerts biological effects in various health and disease conditions.

Additionally, the translational research paradigm tracks research in a continuum from basic discovery research (T0) to changes in community health (T4). However, the linearity of

progress from T0 to T4 almost exclusively applies only to therapeutic drugs or devices. Many nutraceuticals, like RBAC, which are already widely available off-the-shelves, often lack supporting research, and their impact on community health is unknown. Hence, there is a need to conduct community-based research (T4), such as prevalence estimates of use with or without any translational science based on practice (T3), to inform public health decision marking [159]. Hence, assessing the impact of RBAC on community health from the perspective of current or past users may be a topic for future research.

The decision to include conference abstracts is also contentious, even though it is generally a good practice in a systematic review to consider grey literature such as conference abstracts [30, 36, 166]. Proponents of conference abstract inclusion argue that positive results are preferentially published and published sooner, as full-length articles may lag [166]. However, information from the conference abstract may not be reliable as there may be insufficient detail and a lack of peer-reviewed [167]. This study included a total of 8 conference abstracts. Most (5/8) reported early results or preliminary studies, which the authors subsequently worked on and reported further findings and related studies [74, 75, 125, 134, 135, 138]. Two conference abstracts reported positive results from preliminary experiments but no subsequent follow-up studies [124, 146]. One recent conference abstract reported negative results from an RCT, which may be unlikely to receive favourable acceptance for publication as a full-text article [153]. As this study is a scoping review intending to map the translational research on RBAC, we found the need to include all original research done in the field over time, including conference abstracts, to avoid publication bias. The early reporting of results in conference abstracts enables the mapping of specific research keywords to be detected earlier before their follow-up studies were published. For instance, the main findings of RBAC's beneficial effect on liver injury were published by Zhang et al. in 2012 [48, 139]. The preliminary works had started over a decade earlier [134, 138]. Notably, including results from initial experiments that lacked follow-up studies also offered insights into potential further research, for instance, the effects of RBAC on prostate cancer cell line [124] or asthmatic mice model [146].

## Conclusion

RBAC was defined as any rice bran arabinoxylan/polysaccharide extract produced through bioconversion with *L. edodes* mycelial enzyme. This scoping review found 98 primary research articles that investigated RBAC's effects on health and disease conditions published over 25 years from 1998 to 2022. These sources of evidence range from basic, preclinical research to interventional clinical trials in humans, predominantly based on Biobran MGN-3. Research evidence supports RBAC as an immune-modulating nutraceutical for health and disease prevention. RBAC also possesses synergistic anticancer actions that may potentially improve treatment outcomes and the psychoneurological well-being of cancer patients. RBAC has also been studied for potential applications in other conditions, including liver diseases, HIV, common cold/flu, CFS, IBS, etc. However, only four clinical trials are considered T2 research studying the effects of translation on patients. Hence, more translation research on patients and community health is needed to strengthen the evidence base of RBAC for a better understanding of its potential uses and impacts as a nutraceutical.

## Supporting information

**S1 File. Supporting materials for methods.**
(PDF)

**S2 File. PRISMA-ScR checklist.**
(PDF)

**S3 File. Data tables.**
(PDF)

**S4 File. Additional tables and figures.**
(PDF)

**S5 File. Additional references.**
(PDF)

## Acknowledgments

The authors thank Ryo Ninomiya from Daiwa and Seong Gil Hong from Erom, who contributed to searching and sourcing selected Japanese and Korean articles, respectively. We also acknowledge Charles Sturt University for the support of open access publication under the Tri-faculty Open Access Publishing Scheme 2023. These individuals and organisations had no role in the implementation, analyses, data interpretation, or decision to submit study results.

## Author Contributions

**Conceptualization:** Soo Liang Ooi, Peter S. Micalos, Sok Cheon Pak.

**Data curation:** Soo Liang Ooi.

**Investigation:** Soo Liang Ooi.

**Methodology:** Soo Liang Ooi, Peter S. Micalos, Sok Cheon Pak.

**Resources:** Sok Cheon Pak.

**Software:** Soo Liang Ooi.

**Supervision:** Sok Cheon Pak.

**Validation:** Peter S. Micalos, Sok Cheon Pak.

**Visualization:** Soo Liang Ooi.

**Writing – original draft:** Soo Liang Ooi.

**Writing – review & editing:** Soo Liang Ooi, Peter S. Micalos, Sok Cheon Pak.

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
