## [Decision Letter · Decision Letter 0]

5 Jun 2023

PONE-D-23-12330Modified Rice Bran Arabinoxylan as a Nutraceutical in Health and Disease – A Scoping Review with Bibliometric AnalysisPLOS ONE

Dear Dr. Ooi,

Thank you for submitting your manuscript to PLOS ONE. After careful consideration, we feel that it has merit but does not fully meet PLOS ONE’s publication criteria as it currently stands. Therefore, we invite you to submit a revised version of the manuscript that addresses the points raised during the review process.

We look forward to receiving your revised manuscript.

Kind regards,

Dharmendra Kumar Meena

Academic Editor

PLOS ONE

Additional Editor Comments:

The article in its present from have many issue related to to discussion, grammatical and phrases mistakes so i recommends its through revision and recommends its major revision.

Reviewers' comments:

Reviewer's Responses to Questions

**Comments to the Author**

1. Is the manuscript technically sound, and do the data support the conclusions?

Reviewer #1: Yes

Reviewer #2: Yes

2. Has the statistical analysis been performed appropriately and rigorously? 

Reviewer #1: N/A

Reviewer #2: Yes

3. Have the authors made all data underlying the findings in their manuscript fully available?

Reviewer #1: Yes

Reviewer #2: Yes

4. Is the manuscript presented in an intelligible fashion and written in standard English?

Reviewer #1: Yes

Reviewer #2: Yes

5. Review Comments to the Author

Reviewer #1: Reviewer comments

In this work, Ooi et al. conducted a scoping review on the potential health-promoting effects of modified rice bran arabinoxylan as a nutraceutical ingredient in disease management. Kindly, find below my comments for your perusal.

Abstract

The authors should indicate the “databases” where the article search was carried out and the date for article search. In addition, could the authors indicate at least two of the countries that recorded the highest number of research on the RBAC? In this statement “Cancer patients reported reduced side effects from chemoradiotherapy and improved quality of life in human studies………….”, was it when the RBAC was combined with the chemoradiotherapy as an adjuvant? How was the RBAC administered? Was the administered before the intervention? This has to be made clear.

Keyword: The authors should kindly add “health effects” and “physiological effects”.

Introduction

The authors should kindly highlight the global production capacity of rice bran and how its disposal is a nuisance on the environment and thus the need for valorisation into products including RBAC. A brief insight on how the RBAC is produced from the rice bran will be great.

Line 15-19: The authors should state clearly in what form the RBAC is administered. As a supplement? Also, is it only in cancer treatment the RBAC has been used in past research? Other applications in disease management should be highlighted.

Line 16: The authors indicate “Previous reviews…….” Yet they cited just one reference. This is woefully inadequate reference to support the statement.

Methods

Under the section “S1. Supporting Materials for Methods”, “S1.1 Search strategy example”, I couldn’t see “health effect* OR physiological effect*” as part of the search strategy listed below. The information there only touches on the “RBAC” and not any of the health effects associated with its use. Could the authors clarify that?

The authors should also state that the selected articles and its information should was presented using a PRISMA guideline as suggested by Page et al. “Page, M. J., McKenzie, J. E., Bossuyt, P. M., Boutron, I., Hoffmann, T. C., Mulrow, C. D., ... & Moher, D. (2021). The PRISMA 2020 statement: an updated guideline for reporting systematic reviews. International journal of surgery, 88, 105906.”

Eligibility

Line 46: Regarding the statement “…….or the brand/product name…”, do the authors mean that they focused on only commercially produced RBAC? What about RBAC produced on small scale and didn’t have a BRAND name? Why did the authors exclude those studies?

Line 53-54: Why would the authors include articles published as “Conference proceedings” when in most cases, they don’t go through rigorous peer review to ascertain the validity and credibility of the study data?

Information sources

Line 74: Why did the authors include unpublished articles? How would you trust the quality of the data?

Discussion

Line 471-472: Kindly, provide references to support the statement

Line 486: Kindly, provide references to support those statements

Conflict of interest statement

The authors have acknowledged two individuals at the “Acknowledgement” section that works for these commercial entities that are at the forefront of pushing BRAC for consumer purchase. There could be potential conflict of interest there. The authors have to declare that.

Other general comments

1. The authors failed to support most of the statements made under the “Discussion” section with references. This makes the “Discussion” become almost like the “Results” and consequently weakens the Discussion of the review. The authors should kindly discuss this in the context of what others have reported and have done.

2. In the “Results” presentation for example, the authors showed that most of the publication output were recorded from the United States of America, followed closely by Japan, South Korea, Egypt etc. However, the authors failed to discuss this trend in the Discussion. Why was this observed? Could it be attributed to the fact that USA is a leading rice producer and thus generates huge amounts of the rice bran? These have to be highlighted.

3. Even though the authors highlighted the potential health-promoting effects of the BRAC, they failed to indicate the potential mechanism of action and which potential individual compounds in the BRAC that could be driving those physiological effects with the beneficial health-promoting effects observed consequently.

4. The authors should also have assessed the quality of articles that were selected for this review.

5. In fact, if you peruse the total number of references at the Bibliographic references, only 29 references have been indicated. This is unacceptably too small considering the magnitude of this review. The authors should work on this. I find that the authors have additional references, which have been uploaded as part of the supporting information (S4. Additional tables and figures). That Table should be part of the review as it highlights the summary of the study findings.

6. The authors could strengthen the “Discussion” by systematically discussing each of the “Section titles” under the “Results” section. For example “Institutions” etc. The authors should attribute reasons to the observed trends.

Reviewer #2: The effort in this paper is excellent at providing an overview of researches conducted in different countries and published in different languages about the health benefits of Rice Bran Arabinoxylan compound (RBAC). This information is very essential since it gives basic and in fact well-organized information (by identifying clear gaps in the research area) for researchers who want to pursue their research in the field of Nutraceutical research or specifically on RBAC translational research.

I thus recommend the manuscript to be accepted as it is.

6. PLOS authors have the option to publish the peer review history of their article (what does this mean?). If published, this will include your full peer review and any attached files.

Reviewer #1: No

Reviewer #2: No

---

## [Author Response · Author response to Decision Letter 0]

4 Jul 2023

Response to Reviewer 1

We thank you for the constructive feedback. Below is the itemised response to each of the suggestions.

# Comment/Question Response

1. Abstract

The authors should indicate the “databases” where the article search was carried out and the date for article search. In addition, could the authors indicate at least two of the countries that recorded the highest number of research on the RBAC?

Response: Thank you, the suggestions have been incorporated in the updated Abstract (lines 3-5 and 11-12).

2. In this statement “Cancer patients reported reduced side effects from chemoradiotherapy and improved quality of life in human studies………….”, was it when the RBAC was combined with the chemoradiotherapy as an adjuvant? How was the RBAC administered? Was the administered before the intervention? This has to be made clear. 

Response: We have updated the statement: “As an oral supplement taken as an adjuvant during chemoradiotherapy, cancer patients reported reduced side effects and improved quality of life in human studies….” (lines 15-16). 

3 Keyword: The authors should kindly add “health effects” and “physiological effects”.

Response: The suggested keywords are added (see title page). 

4 The authors should kindly highlight the global production capacity of rice bran and how its disposal is a nuisance on the environment and thus the need for valorisation into products including RBAC. A brief insight on how the RBAC is produced from the rice bran will be great.

Response: We have added a new introductory paragraph at the beginning of the manuscript (1st paragraph). More details on how RBAC is typically produced have also been added (lines 14-17).

5 Line 15-19: The authors should state clearly in what form the RBAC is administered. As a supplement? Also, is it only in cancer treatment the RBAC has been used in past research? Other applications in disease management should be highlighted. 

Response: We have highlighted that RBAC was promoted as an adjuvant oral supplement in cancer and also highlighted potential clinical applications in other diseases (lines 28-32). 

6 Line 16: The authors indicate “Previous reviews…….” Yet they cited just one reference. This is woefully inadequate reference to support the statement. 

Response: More references to other reviews have been added (lines 33-35).

7 Under the section “S1. Supporting Materials for Methods”, “S1.1 Search strategy example”, I couldn’t see “health effect* OR physiological effect*” as part of the search strategy listed below. The information there only touches on the “RBAC” and not any of the health effects associated with its use. Could the authors clarify that? 

Response: Yes. The search strategy was intentionally broad as we wished to maximise the search scope to locate all articles related to RBAC. Health and physiological effects were not added as a search term but were naturally captured under the broad search strategy. The total results obtained after the search remained manageable for screening. Hence, additional filtering was not needed.

8 The authors should also state that the selected articles and its information should was presented using a PRISMA guideline as suggested by Page et al. “Page, M. J., McKenzie, J. E., Bossuyt, P. M., Boutron, I., Hoffmann, T. C., Mulrow, C. D., ... & Moher, D. (2021). The PRISMA 2020 statement: an updated guideline for reporting systematic reviews. International journal of surgery, 88, 105906.”

Response: The PRISMA 2020 guideline by Page et al. and specifically the extension on Scoping Review has been cited (lines 151-153). 

9 Line 46: Regarding the statement “…….or the brand/product name…”, do the authors mean that they focused on only commercially produced RBAC? What about RBAC produced on small scale and didn’t have a BRAND name? Why did the authors exclude those studies? 

Response: We have rewritten the statement for better clarity as follows:

“To ensure that only studies on RBAC were selected, an article must contain either (1) an explanation of or references to how the extraction was performed or (2) the brand/product name or provider of the RBAC used. Failure to do so would result in exclusion.” (lines 61-64)

Hence, the selection criteria did not focus only on commercially produced RBAC. Any form of RBAC made on a small scale or in the lab could be included as long as the extraction process involved using L. edodes enzyme in the bioconversion of rice bran. 

10 Line 53-54: Why would the authors include articles published as “Conference proceedings” when in most cases, they don’t go through rigorous peer review to ascertain the validity and credibility of the study data? 

Response: As this study is a scoping review with the intention of mapping the transition research on RBAC, we found the need to include all original research done in the field, including conference abstracts, to avoid publication bias. Conference abstracts with the results not published in full-text articles were included as we found they usually belonged to 3 types: (1) novel findings soon to be published, (2) preliminary results that need follow-up studies, and (3) negative results. 

Hence, when considering the full scope of a research field, we believe including conference abstracts with the results not published in full-text articles could offer insights into gaps in research, potential limitation, or areas worthy of follow-up research.

We have included a paragraph in the discussion section devoted to the rationale of including conference abstracts and their implication to the current study (lines 661-679).

11 Information sources

Line 74: Why did the authors include unpublished articles? How would you trust the quality of the data?

Response: Uncovering unpublished studies/grey literature is also recommended in systematic searches (Paez, 2017) to ensure that all original research within the domain can be located (lines 94-96). To ensure a certain level of quality, we have set the criteria to accept only scholarly articles reporting results from primary research and excluded any non-scholarly sources of magazines, news articles, and trade journals. 

Paez, A. (2017). Gray literature: An important resource in systematic reviews. Journal of Evidence-Based Medicine, 10(3), 233-240. https://doi.org/https://doi.org/10.1111/jebm.12266

12 Discussion

Line 471-472: Kindly, provide references to support the statement

Response: We have updated the statement reflecting that it is a reiteration of our findings from Table 10 (lines 550-552) and provided make references support from other reviews (line 553). 

13 Line 486: Kindly, provide references to support those statements

Response: We have added the requested references (lines 589-590) .

14 Conflict of interest statement

The authors have acknowledged two individuals at the “Acknowledgement” section that works for these commercial entities that are at the forefront of pushing BRAC for consumer purchase. There could be potential conflict of interest there. The authors have to declare that.

Response: We have clarified in the acknowledgement section that “these individuals and their affiliated organisations had no role in the implementation, analyses, data interpretation, or decision to submit results of the study.” (lines 695-698)

15 Other general comments

1. The authors failed to support most of the statements made under the “Discussion” section with references. This makes the “Discussion” become almost like the “Results” and consequently weakens the Discussion of the review. The authors should kindly discuss this in the context of what others have reported and have done. 

Response: We have added all the references to support our discussion. We have also added supporting details from other reviews to strengthen our arguments in the discussion section (page 28-36).

16 

2. In the “Results” presentation for example, the authors showed that most of the publication output were recorded from the United States of America, followed closely by Japan, South Korea, Egypt etc. However, the authors failed to discuss this trend in the Discussion. Why was this observed? Could it be attributed to the fact that USA is a leading rice producer and thus generates huge amounts of the rice bran? These have to be highlighted.

Response: Thank you for this comment; We have added a paragraph discussing the influence of countries on the research output in RBAC (lines 492-506, lines 507-522). 

17 3. Even though the authors highlighted the potential health-promoting effects of the BRAC, they failed to indicate the potential mechanism of action and which potential individual compounds in the BRAC that could be driving those physiological effects with the beneficial health-promoting effects observed consequently.

Response: We have added more discussion on the potential mechanisms of action that could be driving RBAC’s beneficial health-promoting effects observed (lines 550-588). We acknowledge that a detailed review of the mechanisms of action of RBAC is warranted. However, the detailed analysis will be beyond the present manuscript, and we suggest it as an area for follow-up studies in the discussion section (lines 632-634). 

18 4. The authors should also have assessed the quality of articles that were selected for this review.

Response: We have added the reasons for not conducting the requested quality assessment in the discussion (lines 635-651). We agree that quality assessment of evidence is important, but it should be done not at scoping review stage but in a systematic review and meta-analysis which answers a specific clinical question (in PICO form). Hence, we beg to defer the quality assessment for further analysis of the topic (lines 648-651). 

19 5. In fact, if you peruse the total number of references at the Bibliographic references, only 29 references have been indicated. This is unacceptably too small considering the magnitude of this review. The authors should work on this. I find that the authors have additional references, which have been uploaded as part of the supporting information (S4. Additional tables and figures). That Table should be part of the review as it highlights the summary of the study findings.

Response: We have added more of the included studies as references (pages 38-48). However, to avoid cluttering the main text, we choose to leave table S4.1 as a supplementary table. 

20 6. The authors could strengthen the “Discussion” by systematically discussing each of the “Section titles” under the “Results” section. For example “Institutions” etc. The authors should attribute reasons to the observed trends. 

Response: Thank you. We have strengthened the discussion extensively based on your suggestion (pages 28-36).

---

## [Decision Letter · Decision Letter 1]

31 Jul 2023

PONE-D-23-12330R1Modified Rice Bran Arabinoxylan as a Nutraceutical in Health and Disease – A Scoping Review with Bibliometric AnalysisPLOS ONE

Dear Dr. Ooi,

Thank you for submitting your manuscript to PLOS ONE. After careful consideration, we feel that it has merit but does not fully meet PLOS ONE’s publication criteria as it currently stands. Therefore, we invite you to submit a revised version of the manuscript that addresses the points raised during the review process.

We look forward to receiving your revised manuscript.

Kind regards,

Dharmendra Kumar Meena

Academic Editor

PLOS ONE

Additional Editor Comments:

the article is still lacking the cohesion between results a and discussion and as indicated by one the reviewer regarding the ethical issue those needs to be resolved . I encourage its resubmission as major revisions.

Reviewers' comments:

Reviewer's Responses to Questions

**Comments to the Author**

1. If the authors have adequately addressed your comments raised in a previous round of review and you feel that this manuscript is now acceptable for publication, you may indicate that here to bypass the “Comments to the Author” section, enter your conflict of interest statement in the “Confidential to Editor” section, and submit your "Accept" recommendation.

Reviewer #1: All comments have been addressed

2. Is the manuscript technically sound, and do the data support the conclusions?

Reviewer #1: Yes

3. Has the statistical analysis been performed appropriately and rigorously? 

Reviewer #1: N/A

4. Have the authors made all data underlying the findings in their manuscript fully available?

Reviewer #1: Yes

5. Is the manuscript presented in an intelligible fashion and written in standard English?

Reviewer #1: Yes

6. Review Comments to the Author

Reviewer #1: The authors have addressed all the comments I made. I can consequently see that the manuscript has improved significantly in quality.

Well done to the authors.

7. PLOS authors have the option to publish the peer review history of their article (what does this mean?). If published, this will include your full peer review and any attached files.

Reviewer #1: No

---

## [Author Response · Author response to Decision Letter 1]

2 Aug 2023

We appreciate the reviewer’s effort in thoroughly inspecting our work and providing valuable input to improve this manuscript. Thank you very much.

We wish to inform the editor that all reviewers have accepted our manuscript with no additional comments as per the feedback we received as follows:

“Reviewer #1: The authors have addressed all the comments I made. I can consequently see that the manuscript has improved significantly in quality. Well done to the authors.”

As such, there is no outstanding comment to address. We are submitting the manuscript as it is with no change.

I appreciate your support and look forward to hearing your final decision.

---

## [Editor Report · Decision Letter 2]

7 Aug 2023

Modified Rice Bran Arabinoxylan as a Nutraceutical in Health and Disease – A Scoping Review with Bibliometric Analysis

PONE-D-23-12330R2

Dear Dr. Soo

We’re pleased to inform you that your manuscript has been judged scientifically suitable for publication and will be formally accepted for publication once it meets all outstanding technical requirements.

Kind regards,

Dharmendra Kumar Meena

Academic Editor

PLOS ONE
---

## [Editor Report · Acceptance letter]

22 Aug 2023

PONE-D-23-12330R2 

Modified Rice Bran Arabinoxylan as a Nutraceutical in Health and Disease – A Scoping Review with Bibliometric Analysis 

Dear Dr. Ooi:

I'm pleased to inform you that your manuscript has been deemed suitable for publication in PLOS ONE. Congratulations! Your manuscript is now with our production department. 

Kind regards, 

on behalf of

Dr. Dharmendra Kumar Meena 

Academic Editor

PLOS ONE